# Bypassing the Ambient Dimension: Private SGD with Gradient Subspace Identification

**Yingxue Zhou**[†], **Zhiwei Steven Wu**[‡], **Arindam Banerjee**[§]

[†] Department of Computer Science & Engineering, University of Minnesota
[‡] School of Computer Science, Carnegie Mellon University.
[§] Department of Computer Science, University of Illinois Urbana-Champaign
[†] zhou0877@umn.edu, [‡] zstevenwu@cmu.edu, [§] arindamb@illinois.edu

## Abstract

Differentially private SGD (DP-SGD) is one of the most popular methods for solving differentially private empirical risk minimization (ERM). Due to its noisy perturbation on each gradient update, the error rate of DP-SGD scales with the ambient dimension $p$, the number of parameters in the model. Such dependence can be problematic for over-parameterized models where $p \gg n$, the number of training samples. Existing lower bounds on private ERM show that such dependence on $p$ is inevitable in the worst case. In this paper, we circumvent the dependence on the ambient dimension by leveraging a low-dimensional structure of gradient space in deep networks—that is, the stochastic gradients for deep nets usually stay in a low dimensional subspace in the training process. We propose Projected DP-SGD that performs noise reduction by projecting the noisy gradients to a low-dimensional subspace, which is given by the top gradient eigenspace on a small public dataset. We provide a general sample complexity analysis on the public dataset for the gradient subspace identification problem and demonstrate that under certain low-dimensional assumptions the public sample complexity only grows logarithmically in $p$. Finally, we provide a theoretical analysis and empirical evaluations to show that our method can substantially improve the accuracy of DP-SGD in the high privacy regime (corresponding to low privacy loss $\epsilon$).

## 1 Introduction

Many fundamental machine learning tasks involve solving empirical risk minimization (ERM): given a loss function $\ell$, find a model $\mathbf{w} \in \mathbb{R}^p$ that minimizes the *empirical risk* $\hat{L}_n(\mathbf{w}) = \frac{1}{n} \sum_{i=1}^{n} \ell(\mathbf{w}, z_i)$, where $z_1, \ldots, z_n$ are i.i.d. examples drawn from a distribution $\mathcal{P}$. In many applications, the training data may contain highly sensitive information about some individuals. When the models are given by deep neural networks, their rich representation can potentially reveal fine details of the private data.

*Differential privacy* (DP) (Dwork et al., 2006) has by now become the standard approach to provide principled and rigorous privacy guarantees in machine learning. Roughly speaking, DP is a stability notion that requires that no individual example has a significant influence on the trained model. One of the most commonly used algorithm for solving private ERM is the *differentially-private stochastic gradient descent* (DP-SGD) (Abadi et al., 2016; Bassily et al., 2014; Song et al., 2013)–a private variant of SGD that perturbs each gradient update with random noise vector drawn from an isotropic Gaussian distribution $\mathcal{N}(\mathbf{0}, \sigma^2 \mathbb{I}_p)$, with appropriately chosen variance $\sigma^2$.

Due to the gradient perturbation drawn from an isotropic Gaussian distribution, the error rate of DP-SGD has a dependence on the ambient dimension $p$—the number of parameters in the model. In the case of convex loss $\ell$, Bassily et al. (2014) show that DP-SGD achieves the optimal empirical excess risk of $\tilde{O}\left(\sqrt{p}/(n\epsilon)\right)$. For non-convex loss $\ell$, which is more common in neural network training, minimizing $\hat{L}_n(\mathbf{w})$ is in general intractable. However, many (non-private) gradient-based optimization methods are shown to be effective in practice and can provably find approximate stationary points with vanishing gradient norm $\|\nabla \hat{L}_n(\mathbf{w})\|_2$ (see e.g. Nesterov (2014); Ghadimi and

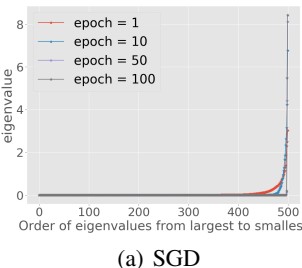 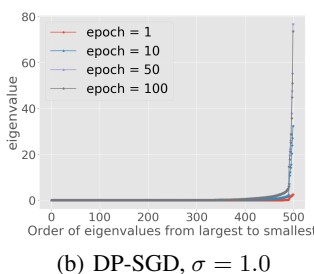 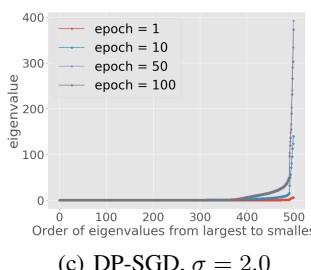

(a) SGD      (b) DP-SGD, $\sigma = 1.0$      (c) DP-SGD, $\sigma = 2.0$

Figure 1: Top 500 eigen-value spectrum of the gradient second moment matrix along the training trajectory of SGD, DP-SGD with $\sigma = 1, 2$. Dataset: MNIST; model: 2-layer ReLU with 128 nodes each layer. The network has roughly 130,000 parameters and is trained on MNIST dataset. The Y-axis is the eigenvalue and X-axis is order of eigenvalues from largest to smallest.

Lan (2013)). Moreover, for a wide family of loss functions $\hat{L}_n$ under the Polyak-Łojasiewicz condition (Polyak, 1963), the minimization of gradient norm implies achieving global optimum. With privacy constraint, Wang and Xu (2019) recently showed that DP-SGD minimize the empirical gradient norm down to $\tilde{O}\left(p^{1/4}/\sqrt{n\epsilon}\right)$ when the loss function $\ell$ is smooth. Furthermore, exsiting lower bounds results on private ERM (Bassily et al., 2014) show that such dependence on $p$ is inevitable in the worst case. However, many modern machine learning tasks now involve training extremely large models, with the number of parameters substantially larger than the number of training samples. For these large models, the error dependence on $p$ can be a barrier to practical private ERM.

In this paper, we aim to overcome such dependence on the ambient dimension $p$ by leveraging the structure of the gradient space in the training of neural networks. We take inspiration from the empirical observation from Li et al. (2020); Gur-Ari et al. (2018); Papyan (2019) that even though the ambient dimension of the gradients is large, the set of sample gradients at most iterations along the optimization trajectory is often contained in a much lower-dimensional subspace. While this observation has been made mostly for non-private SGD algorithm, we also provide our empirical evaluation of this structure (in terms of eigenvalues of the gradient second moments matrix) in Figure 1. Based on this observation, we provide a modular private ERM optimization framework with two components. At each iteration $t$, the algorithm performs the following two steps:

*1) Gradient dimension reduction.* Let $\mathbf{g}_t$ be the mini-batch gradient at iteration $t$. In general, this subroutines solves the following problem: given any $k < p$, find a linear projection $\hat{V}_k(t) \in \mathbb{R}^{p \times k}$ such that the reconstruction error $\|\mathbf{g}_t - \hat{V}_k(t)\hat{V}_k(t)^\intercal \mathbf{g}_t\|$ is small. To implement this subroutine, we follow a long line of work that studies private data analysis with access to an auxiliary public dataset $S_h$ drawn from the same distribution $\mathcal{P}$, for which we don't need to provide formal privacy guarantee (Bassily et al., 2019b; 2020; Feldman et al., 2018; Avent et al., 2017; Papernot et al., 2017). In our case, we compute $\hat{V}_k(t)$ which is given by the top-$k$ eigenspace of the gradients evaluated on $S_h$. Alternatively, this subroutine can potentially be implemented through private subspace identification on the private dataset. However, to our best knowledge, all existing methods have reconstruction error scaling with $\sqrt{p}$ (Dwork et al., 2014), which will be propagated to the optimization error.

*2) Projected DP-SGD (PDP-SGD).* Given the projection $\hat{V}_k(t)$, we perturb gradient in the projected subspace: $\tilde{\mathbf{g}}_t = \hat{V}_k(t)\hat{V}_k(t)^\intercal(\mathbf{g}_t + \mathbf{b}^t)$, where $\mathbf{b}^t$ is a $p$-dimensional Gaussian vector. The projection mapping provides a large reduction of the noise and enables higher accuracy for PDP-SGD.

**Our results.** We provide both theoretical analyses and empirical evaluations of PDP-SGD:

*Uniform convergence for projections.* A key step in our theoretical analysis is to bound the reconstruction error on the gradients from projection of $\hat{V}_k(t)$. This reduces to bounding the deviation $\|\hat{V}_k(t)\hat{V}_k(t)^\intercal - V_k(t)V_k(t)^\intercal\|_2$, where $V_k(t)$ denotes the top-$k$ eigenspace of the population second moment matrix $\mathbb{E}[\nabla \ell(\mathbf{w}_t, z)\nabla \ell(\mathbf{w}_t, z)^\intercal]$. To handle the adaptivity of the sequence of iterates, we provide a uniform deviation bound for all $\mathbf{w} \in \mathcal{W}$, where the set $\mathcal{W}$ contains all of the iterates. By leveraging generic chaining techniques, we provide a deviation bound that scales linearly with a complexity measure—the $\gamma_2$ function due to Talagrand (2014)—of the set $\mathcal{W}$. We provide low-complexity examples of $\mathcal{W}$ that are supported by empirical observations and show that their $\gamma_2$ function only scales logarithmically with $p$.

*Convergence for convex and non-convex optimization.* Building on the reconstruction error bound, we provide convergence and sample complexity results for our method PDP-SGD in two types of loss functions, including 1) smooth and non-convex, 2) Lipschitz convex. Under suitable assumptions on the gradient space, our rates only scales logarithmically on $p$.

*Empirical evaluation.* We provide an empirical evaluation of PDP-SGD on two real datasets. In our experiments, we construct the "public" datasets by taking very small random sub-samples of these two datasets (100 samples). While these two public datasets are not sufficient for training an accurate predictor, we demonstrate that they provide useful gradient subspace projection and substantial accuracy improvement over DP-SGD.

**Related work.** Beyond the aforementioned work, there has been recent work on private ERM that also leverages the low-dimensional structure of the problem. Jain and Thakurta (2014); Song et al. (2020) show dimension independent excess empirical risk bounds for convex generalized linear problems, when the input data matrix is low-rank. Kairouz et al. (2020) study unconstrained convex empirical risk minimization and provide a noisy AdaGrad method that achieves dimension-free excess risk bound, provided that the gradients along the optimization trajectory lie in a low-dimensional subspace. In comparison, our work studies both convex and non-convex problems and our analysis applies to more general low-dimensional structures that can be characterized by small $\gamma_2$ functions (Talagrand, 2014; Gunasekar et al., 2015) (e.g., low-rank gradients and fast decay in the magnitude of the gradient coordinates). Recently, Tramer and Boneh (2021) show that private learning with features learned on public data from a similar domain can significantly improve the utility. Zhang et al. (2021) leverage the sparsity of the gradients in deep nets to improve the dependence on dimension in the error rate. We also note a recent work (Yu et al., 2021) that proposes an algorithm similar to PDP-SGD. However, in addition to perturbing the projected gradient in the top eigenspaces in the public data, their algorithm also adds noise to the residual gradient. Their error rate scales with dimension $p$ in general due to the noise added to the full space. To achieve a dimension independent error bound, their analyses require *fresh* public samples drawn from the same distribution at each step, which consequently requires a large public data set with size scaling linearly with $T$. In comparison, our analysis does not require fresh public samples at each iteration, and our experiments demonstrate that a small public data set of size no more than 150 suffices.[1]

## 2 PRELIMINARIES

Given a private dataset $S = \{z_1, ..., z_n\}$ drawn i.i.d. from the underlying distribution $\mathcal{P}$, we want to solve the following empirical risk minimization (ERM) problem subject to differential privacy:[2] $\min_{\mathbf{w}} \hat{L}_n(\mathbf{w}) = \frac{1}{n} \sum_{i=1}^{n} \ell(\mathbf{w}, z_i)$. where the parameter $\mathbf{w} \in \mathbb{R}^p$. We optimize this objective with an iterative algorithm. At each step $t$, we write $\mathbf{w}_t$ as the algorithm's iterate and use $\mathbf{g}_t$ to denote the mini-batch gradient, and $\nabla \hat{L}_n(\mathbf{w}_t) = \frac{1}{n} \sum_{i=1}^{n} \nabla \ell(\mathbf{w}_t, z_i)$ to denote the empirical gradient. In addition to the private dataset, the algorithm can also freely access to a small public dataset $S_h = \{\tilde{z}_1, \ldots, \tilde{z}_m\}$ drawn from the same distribution $\mathcal{P}$, without any privacy constraint.

**Notation.** We write $M_t \in \mathbb{R}^{p \times p}$ to denote the second moment matrix of gradients evaluated on public dataset $S_h$, i.e., $M_t = \frac{1}{m} \sum_{i=1}^{m} \nabla \ell(\mathbf{w}_t, \tilde{z}_i) \nabla \ell(\mathbf{w}_t, \tilde{z}_i)^{\mathsf{T}}$ and write $\Sigma_t \in \mathbb{R}^{p \times p}$ to denote the population second moment matrix, i.e., $\Sigma_t = \mathbb{E}_{z \sim \mathcal{P}} [\nabla \ell(\mathbf{w}_t, z) \nabla \ell(\mathbf{w}_t, z)^{\mathsf{T}}]$. We use $V(t) \in \mathbb{R}^{p \times p}$ as the full eigenspace of $\Sigma_t$. We use $\hat{V}_k(t) \in \mathbb{R}^{p \times k}$ as the top-$k$ eigenspace of $M_t$ and $V_k(t) \in \mathbb{R}^{p \times k}$ as the top-$k$ eigenspace of $\Sigma_t$. To present our result in the subsequent sections, we introduce the eigen-gap notation $\alpha_t$, i.e., let $\lambda_1(\Sigma_t) \geq ... \geq \lambda_p(\Sigma_t)$ be the eigenvalue of $\Sigma_t$, we use $\alpha_t$ to denote the eigen-gap between $\lambda_k(\Sigma_t)$ and $\lambda_{k+1}(\Sigma_t)$, i.e., $\lambda_k(\Sigma_t) - \lambda_{k+1}(\Sigma_t) \geq \alpha_t$. We also define $\mathcal{W} \in \mathbb{R}^p$ as the set that contains all the possible iterates $\mathbf{w}_t \in \mathcal{W}$ for $t \in [T]$. Throughout, for any matrix $A$ and vector $\mathbf{v}$, $\|A\|_2$ denotes spectral norm and $\|\mathbf{v}\|_2$ denotes $\ell_2$ norm.

**Definition 1 (Differential Privacy (Dwork et al., 2006))** *A randomized algorithm $\mathcal{R}$ is $(\epsilon, \delta)$-differentially private if for any pair of datasets $D, D'$ differ in exactly one data point and for all event*

---

[1]Note that the requirement of a large public data set may remove the need of using the private data in the first place, since training with the large public data set may already provide an accurate model.

[2]In this paper, we focus on minimizing the empirical risk. However, by relying on the generalization guarantee of $(\epsilon, \delta)$-differential privacy, one can also derive a population risk bound that matches the empirical risk bound up to a term of order $O(\epsilon + \delta)$ (Dwork et al., 2015; Bassily et al., 2016; Jung et al., 2020).

$\mathcal{Y} \subseteq Range(\mathcal{R})$ in the output range of $\mathcal{R}$, we have $P\{\mathcal{R}(D) \in \mathcal{Y}\} \leq \exp(\epsilon)P\{\mathcal{R}(D') \in \mathcal{Y}\} + \delta$, where the probability is taken over the randomness of $\mathcal{R}$.

To establish the privacy guarantee of our algorithm, we will combine three standard tools in differential privacy, including 1) the Gaussian mechanism (Dwork et al., 2006) that releases an aggregate statistic (e.g., the empirical average gradient) by Gaussian perturbation, 2) privacy amplification via subsampling (Kasiviswanathan et al., 2008) that reduces the privacy parameters $\epsilon$ and $\delta$ by running the private computation on a random subsample, and 3) advanced composition theorem (Dwork et al., 2010) that tracks the cumulative privacy loss over the course of the algorithm.

We analyze our method under two asumptions on the gradients of $\ell$.

**Assumption 1** *For any $\mathbf{w} \in \mathbb{R}^p$ and example $z$, $\|\nabla \ell(\mathbf{w}, z)\|_2 \leq G$.*

**Assumption 2** *For any example $z$, the gradient $\nabla \ell(\mathbf{w}, z)$ is $\rho$-Lipschitz with respect to a suitable pseudo-metric $d : \mathbb{R}^p \times \mathbb{R}^p \mapsto \mathbb{R}$, i.e., $\|\nabla \ell(\mathbf{w}, z) - \nabla \ell(\mathbf{w}', z)\|_2 \leq \rho d(\mathbf{w}, \mathbf{w}')$, $\forall \mathbf{w}, \mathbf{w}' \in \mathbb{R}^p$.*

Note that Assumption 1 implies that $\hat{L}_n(\mathbf{w})$ is $G$-Lipschitz and Assumption 2 implies that $\hat{L}_n(\mathbf{w})$ is $\rho$-smooth when $d$ is the $\ell_2$-distance. We will discuss additional assumptions regarding the structure of the stochastic gradients and the error rate for different type of functions in Section 3.

## 3 PROJECTED PRIVATE GRADIENT DESCENT

The PDP-SGD follows the classical noisy gradient descent algorithm DP-SGD (Wang et al., 2017; Wang and Xu, 2019; Bassily et al., 2014). DP-SGD adds isotropic Gaussian noise $\mathbf{b}_t \sim \mathcal{N}(0, \sigma^2 \mathbb{I}_p)$ to the gradient $\mathbf{g}_t$, i.e., each coordinate of the gradient $\mathbf{g}_t$ is perturbed by the Gaussian noise. Given the dimension of gradient to be $p$, this method ends up in getting a factor of $p$ in the error rate (Bassily et al., 2014; 2019a). Our algorithm is inspired by the recent observations that stochastic gradients stay in a low-dimensional space in the training of deep nets (Li et al., 2020; Gur-Ari et al., 2018). Such observation is also valid for the private training algorithm, i.e., DP-SGD (Figure 1 (b) and (c)). Intuitively, the most information needed for gradient descent is embedded in the top eigenspace of the stochastic gradients. Thus, PDP-SGD performs noise reduction by projecting the noisy gradient $\mathbf{g}_t + \mathbf{b}_t$ to an approximation of such a subspace given by a public dataset $S_h$.

---

**Algorithm 1** Projected DP-SGD (PDP-SGD)

---

1: **Input**: Training set $S$, public set $S_h$, certain loss $\ell(\cdot)$, initial point $\mathbf{w}_0$
2: **Set**: Noise parameter $\sigma$, iteration time $T$, step size $\eta_t$.
3: **for** $t = 0, ..., T$ **do**
4:     Compute top-$k$ eigenspace $\hat{V}_k(t)$ of $M_t = \frac{1}{|S_h|} \sum_{\tilde{z}_i \in S_h} \nabla \ell(\mathbf{w}_t, \tilde{z}_i) \nabla \ell(\mathbf{w}_t, \tilde{z}_i)^{\mathsf{T}}$.
5:     $\mathbf{g}_t = \frac{1}{|B_t|} \sum_{z_i \in B_t} \nabla \ell(\mathbf{w}_t, z_i)$, with $B_t$ uniformly sampled from $S$ with replacement.
6:     Project noisy gradient using $\hat{V}_k(t)$: $\tilde{\mathbf{g}}_t = \hat{V}_k(t)\hat{V}_k(t)^{\mathsf{T}}(\mathbf{g}_t + \mathbf{b}_t)$, where $\mathbf{b}_t \sim \mathcal{N}(0, \sigma^2 \mathbb{I}_p)$.
7:     Update parameter using projected noisy gradient: $\mathbf{w}_{t+1} = \mathbf{w}_t - \eta_t \tilde{\mathbf{g}}_t$.
8: **end for**

---

Thus, our algorithm involves two steps at each iteration, i.e., subspace identification and noisy gradient projection. The pseudo-code of PDP-SGD is given in Algorithm 1. At each iteration $t$, in order to obtain an approximated subspace without leaking the information of the private dataset $S$, we evaluate the second moment matrix $M_t$ on $S_h$ and compute the top-$k$ eigenvectors $\hat{V}_k(t)$ of $M_t$ (line 4 in Algorithm 1). Then we project the noisy gradient $\mathbf{g}_t + \mathbf{b}_t$ to the top-$k$ eigenspace, i.e., $\tilde{\mathbf{g}}_t = \hat{V}_k(t)\hat{V}_k(t)^{\mathsf{T}}(\mathbf{g}_t + \mathbf{b}_t)$ (line 6 in Algorithm 1). Then PDP-SGD uses the projected noisy gradient $\tilde{\mathbf{g}}_t$ to update the parameter $\mathbf{w}_{t+1} = \mathbf{w}_t - \eta_t \tilde{\mathbf{g}}_t$.[3] Let us first state its privacy guarantee.

**Theorem 1 (Privacy)** *Under Assumption 1, there exist constants $c_1$ and $c_2$ so that given the number of iterations $T$, for any $\epsilon \leq c_1 q^2 T$, where $q = \frac{|B_t|}{n}$, PDP-SGD (Algorithm 1) is $(\epsilon, \delta)$-differentially private for any $\delta > 0$, if $\sigma^2 \geq c_2 \frac{G^2 T \ln\left(\frac{1}{\delta}\right)}{n^2 \epsilon^2}$.*

---

[3]For convex problem, we consider the typical constrained optimization problem such that the optimal solution $\mathbf{w}^\star$ is in a set $\mathcal{H}$, where $\mathcal{H} = \{\mathbf{w} : \|\mathbf{w}\| \leq B\}$ and each step we project $\mathbf{w}_{t+1}$ back to the set $\mathcal{H}$.

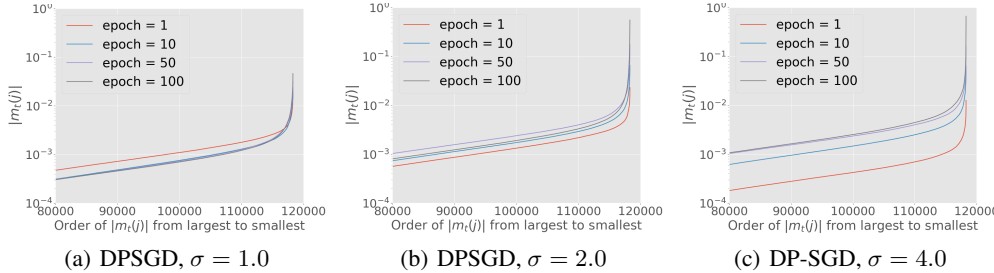

(a) DPSGD, $\sigma = 1.0$      (b) DPSGD, $\sigma = 2.0$      (c) DP-SGD, $\sigma = 4.0$

Figure 2: Sorted components of population gradients for DP-SGD with $\sigma = 1.0,\ 2.0,\ 4.0$. Dataset: MNIST; model: 2-layer ReLU with 128 nodes each layer. The network has roughly 130,000 parameters. Y-axis is the absolute value of sorted gradient coordinates, i.e., $|\mathbf{m}_t(j)|$, X-axis is the order of sorted gradient component.

The privacy proof essentailly follows from the same proof of DP-SGD (Abadi et al., 2016). At each iteration, the update step PDP-SGD is essentially post-processing of Gaussian Mechanism that computes a noisy estimate of the gradient $\mathbf{g}_t + \mathbf{b}_t$. Then the privacy guarantee of releasing the sequence of $\{\mathbf{g}_t + \mathbf{b}_t\}_t$ is exactly the same as the privacy proof of Theorem 1 of Abadi et al. (2016).

### 3.1 GRADIENT SUBSPACE IDENTIFICATION

We now analyze the gradient deviation between the approximated subspace $\hat{V}_k(t)\hat{V}_k(t)^\intercal$ and true (population) subspace $V_k(t)V_k(t)^\intercal$, i.e., $\|\hat{V}_k(t)\hat{V}_k(t)^\intercal - V_k(t)V_k(t)^\intercal\|_2$. To bound $\|\hat{V}_k(t)\hat{V}_k(t)^\intercal - V_k(t)V_k(t)^\intercal\|_2$, we first bound the deviation between second moment matrix $\|M_t - \Sigma_t\|$ (Dwork et al., 2014; McSherry, 2004). Note that, if $M_t = \frac{1}{m}\sum_{i=1}^m \nabla\ell(\mathbf{w}_t, \tilde{z}_i)\nabla\ell(\mathbf{w}_t, \tilde{z}_i)^\intercal$ is evaluated on *fresh* public samples, the $\Sigma_t$ is the expectation of $M_t$, and the deviation of $M_t$ from $\Sigma_t$ can be easily analyzed by the Ahlswede-Winter Inequality (Horn and Johnson, 2012; Wainwright, 2019), i.e., at any iteration $t$, if we have fresh public sample $\{\tilde{z}_1(t), ..., \tilde{z}_m(t)\}$ drawn i.i.d. from the distribution $\mathcal{P}$, with suitable assumptions we have, $\left\|\frac{1}{m}\sum_{i=1}^m \nabla\ell(\mathbf{w}_t, \tilde{z}_i(t))\nabla\ell(\mathbf{w}_t, \tilde{z}_i(t))^\intercal - \mathbb{E}\left[\nabla\ell(\mathbf{w}_t, \tilde{z}_i(t))\nabla\ell(\mathbf{w}_t, \tilde{z}_i(t))^\intercal\right]\right\|_2 > u,\ \forall u \in [0, 1]$, with probability at most $p\exp(-mu^2/4G)$ and $G$ is as in Assumption 1.

However, this concentration bound does not hold for $\mathbf{w}_t, \forall t > 0$ in general, since the public dataset $S_h$ is *reused* over the iterations and the parameter $\mathbf{w}_t$ depends on $S_h$. To handle the dependency issue, we bound $\|M_t - \Sigma_t\|_2$ uniformly over all iterations $t \in [T]$ to bound the worst-case counterparts that consider all possible iterates. Our uniform bound analysis is based on generic chaining (GC) (Talagrand, 2014), an advanced tool from probability theory. Eventually, the error bound is expressed in terms of a complexity measure called $\gamma_2$ function (Talagrand, 2014). Note that one may consider the idea of sample splitting to bypass the dependency issue by splitting $m$ public samples into $T$ disjoint subsets for each iteration. Based on Ahlswede-Winter Inequality, the deviation error scales with $O(\frac{\sqrt{T}}{\sqrt{m}})$ leading to a worse trade-off between the subspace construction error and optimization error due to the dependence on $T$.

**Definition 2 ($\gamma_2$ function (Talagrand, 2014))** *For a metric space $(\mathcal{A}, d)$, an admissible sequence of $\mathcal{A}$ is a collection of subsets of $\mathcal{A}$, $\Gamma = \{\mathcal{A}_n : n \geq 0\}$, with $|\mathcal{A}_0| = 1$ and $|\mathcal{A}_n| \leq 2^{2^n}$ for all $n \geq 1$, the $\gamma_2$ functional is defined by $\gamma_2(\mathcal{A}, d) = \inf_\Gamma \sup_{A \in \mathcal{A}} \sum_{n \geq 0} 2^{n/2} d(A, \mathcal{A}_n)$, where the infimum is over all admissible sequences of $\mathcal{A}$.*

In Theorem 2, we show that the uniform convergence bound of $\|M_t - \Sigma_t\|_2$ scales with $\gamma_2(\mathcal{W}, d)$, where $d$ is the pseudo metric as in Assumption 2 and $\mathcal{W} \in \mathbb{R}^p$ is the set that contains all possible iterates in the algorithm, i.e., $\mathbf{w}_t \in \mathcal{W}$ for all $t \in [T]$.

Based on the majorizing measure theorem (e.g., Theorem 2.4.1 in Talagrand (2014)), if the metric $d$ is $\ell_2$-norm, $\gamma_2(\mathcal{W}, d)$ can be expressed as Gaussian width (Vershynin, 2018; Wainwright, 2019) of the set $\mathcal{W}$, i.e., $w(\mathcal{W}) = \mathbb{E}_\mathbf{v}[\sup_{\mathbf{w} \in \mathcal{W}}\langle\mathbf{w}, \mathbf{v}\rangle]$ where $\mathbf{v} \sim \mathcal{N}(0, \mathbb{I}_p)$, which only depends on the size of the $\mathcal{W}$. In Appendix A.2, we show the complexity measure $\gamma_2(\mathcal{W}, d)$ can be expressed as the $\gamma_2$ function measure on the gradient space by mapping the parameter space $\mathcal{W}$ to the gradient space, i.e., $f : \mathcal{W} \mapsto \mathcal{M}$, where $f$ can be considered as $f(\mathbf{w}) = \mathbb{E}_{z \in \mathcal{P}}[\nabla(\mathbf{w}, z)]$. To simplify the

notation, we write $\mathbf{m} = \mathbb{E}_{z \in \mathcal{P}} [\nabla(\mathbf{w}, z)]$ as the population gradient at $\mathbf{w}$ and $\mathcal{M}$ as the space of the population gradient. Considering $d(\mathbf{m}, \mathbf{m}') = \|\mathbf{m} - \mathbf{m}'\|_2$ for $\mathbf{m}, \mathbf{m}' \in \mathcal{M}$, $\gamma_2(\mathcal{M}, d)$ will be the same order as the Gaussian width $w(\mathcal{M})$.

To measure the value of $\gamma_2(\mathcal{M}, d)$, we empirically explore the gradient space $\mathcal{M}$ for deep nets. Figure 2 gives an example of the population gradient along the training trajectory of DP-SGD with $\sigma = \{1, 2, 4\}$ for training a 2-layer ReLU on MNIST dataset. Figure 2 shows that each coordinate of the gradient is of small value and gradient components decay very fast (Li and Banerjee, 2021). Thus, it is fair that the gradient space $\mathcal{M}$ is a union of ellipsoids, i.e., there exists $\mathbf{e} \in \mathbb{R}^p$ such that $\mathcal{M} = \{\mathbf{m} \in \mathbb{R}^p \mid \sum_{j=1}^{p} \mathbf{m}(j)^2 / \mathbf{e}(j)^2 \leq 1, \mathbf{e} \in \mathbb{R}^p\}$, where $j$ denotes the $j$-th coordinate. Then we have $\gamma_2(\mathcal{M}, d) \leq c_1 w(\mathcal{M}) \leq c_2 \|\mathbf{e}\|_2$ (Talagrand, 2014), where $c_1$ and $c_2$ are absolute constants. If the elements of $\mathbf{e}$ are sorted in a decreasing order satisfy $\mathbf{e}(j) \leq c_3 / \sqrt{j}$ for all $j \in [p]$, then $\gamma_2(\mathcal{M}, d) \leq O\left(\sqrt{\log p}\right)$.[4] Now we give the uniform convergence bound of $\|M_t - \Sigma_t\|_2$.

**Theorem 2 (Second Moment Concentration)** *Under Assumption 1, 2, the second moment matrix of the public gradient $M_t = \frac{1}{m} \sum_{i=1}^{m} \nabla \ell(\mathbf{w}_t, \tilde{z}_i) \nabla \ell(\mathbf{w}_t, \tilde{z}_i)^\intercal$ approximates the population second moment matrix $\Sigma_t = \mathbb{E}_{z \sim \mathcal{P}}[\nabla \ell(\mathbf{w}_t, z) \nabla \ell(\mathbf{w}_t, z)^\intercal]$ uniformly over all iterations, i.e., for any $u > 0$,*

$$\sup_{t \in [T]} \|M_t - \Sigma_t\|_2 \leq O\left(\frac{uG\rho\sqrt{\ln p}\gamma_2(\mathcal{W}, d)}{\sqrt{m}}\right), \tag{1}$$

*with probability at least $1 - c \exp\left(-u^2/4\right)$, where $c$ is an absolute constant.*

Theorem 2 shows that $M_t$ approximates the population second moment matrix $\Sigma_t$ uniformly over all iterations. This uniform bound is derived by the technique GC, which develops sharp upper bounds to suprema of stochastic processes indexed by a set with a metric structure in terms of $\gamma_2$ functions. In our case, $\|M_t - \Sigma_t\|$ is treated as the stochastic process indexed by the set $\mathcal{W} \in \mathbb{R}^p$ such that $\mathbf{w}_t \in \mathcal{W}$, which is the set of all possible iterates. The metric $d$ is the pseudo-metric $d : \mathbb{R}^p \times \mathbb{R}^p \mapsto \mathbb{R}$ defined in Assumption 2. To get a more practical bound, following the above discussion, instead of working with $\mathcal{W}$ over parameters, one can consider working with the set $\mathcal{M}$ of population gradients by defining the pseudo-metric as $d(\mathbf{w}, \mathbf{w}') = d(f(\mathbf{w}), f(\mathbf{w}')) = d(\mathbf{m}, \mathbf{m}')$, where $f(\mathbf{w}) = \mathbb{E}_{z \in \mathcal{P}} [\nabla(\mathbf{w}, z)]$ that maps the parameter space to gradient space. Thus, the complexity measure $\gamma_2(\mathcal{W}, d)$ can be expressed as the $\gamma_2$ function measure on the population gradient space, i.e., $\gamma_2(\mathcal{M}, d)$. As discussed above, using the $\ell_2$-norm as $d$, the $\gamma_2(\mathcal{M}, d)$ will be a constant if assuming the gradient space is a union of ellipsoids and uniform bound only depends on logarithmically on $p$.

Using the result in Theorem 2 and Davis-Kahan sin-$\theta$ theorem (McSherry, 2004), we obtain the subspace construction error $\|\hat{V}_k(t)\hat{V}_k(t)^\intercal - V_k(t)V_k(t)^\intercal\|_2$ in the following theorem.

**Theorem 3 (Subspace Closeness)** *Under Assumption 1 and 2, with $V_k(t)$ to be the top-$k$ eigenvectors of the population second moment matrix $\Sigma_t$ and $\alpha_t$ be the eigen-gap at $t$-th iterate such that $\lambda_k(\Sigma_t) - \lambda_{k+1}(\Sigma_t) \geq \alpha_t$, for the $\hat{V}_k(t)$ in Algorithm 1, if $m \geq \frac{O\left(G\rho\sqrt{\ln p}\gamma_2(\mathcal{W},d)\right)^2}{\min_t \alpha_t^2}$, we have*

$$\mathbb{E}\left[\|\hat{V}_k(t)\hat{V}_k(t)^\intercal - V_k(t)V_k(t)^\intercal\|_2\right] \leq O\left(\frac{G\rho\sqrt{\ln p}\gamma_2(\mathcal{W}, d)}{\alpha_t\sqrt{m}}\right), \forall t \in [T]. \tag{2}$$

Theorem 3 gives the sample complexity of the public sample size and the reconstruction error, i.e., the difference between $\hat{V}_k(t)\hat{V}_k(t)^\intercal$ evaluated on the public dataset $S_h$ and $V_k(t)V_k(t)^\intercal$ given by the population second moment $\Sigma_t$. The sample complexity and the reconstruction error both depend on the $\gamma_2$ function and eigen-gap $\alpha_t$. A small eigen-gap $\alpha_t$ requires larger public sample $m$.

## 3.2 EMPIRICAL RISK CONVERGENCE ANALYSIS

In this section, we present the error rate of PDP-SGD for non-convex (smooth) functions. The error rate for convex functions is deferred to Appendix C. For non-convex case, we first give the error rate of the $\ell_2$-norm of the principal component of the gradient, i.e., $\|V_k(t)V_k(t)^\intercal \nabla \hat{L}_n(\mathbf{w}_t)\|_2$. Then we show that the gradient norm also converges if the principal component dominates the residual component of the gradient as suggested by Figure 1 and recent observations (Papyan, 2019; Li et al.,

---

[4]In the appendix, we provide more examples $\mathcal{M}$ that are consistent with empirical observations of stochastic gradient distributions and have small $\gamma_2(\mathcal{M}, d)$.

2020). To present our results, we introduce some new notations here. We write $[\nabla \hat{L}_n(\mathbf{w}_t)]^{\|} = V_k(t)V_k(t)^{\mathsf{T}}\nabla \hat{L}_n(\mathbf{w}_t)$ as the principal component of the gradient and $[\nabla \hat{L}_n(\mathbf{w}_t)]^{\perp} = \nabla \hat{L}_n(\mathbf{w}_t) - V_k(t)V_k(t)^{\mathsf{T}}\nabla \hat{L}_n(\mathbf{w}_t)$ as the residual component.

**Theorem 4 (Smooth and Non-convex)** *For $\rho$-smooth function $\hat{L}_n(\mathbf{w})$, under Assumptions 1 and 2, let $\Lambda = \frac{\sum_{t=1}^{T} 1/\alpha_t^2}{T}$, for any $\epsilon, \delta > 0$, with $T = O(n^2\epsilon^2)$ and $\eta_t = \frac{1}{\sqrt{T}}$, PDP-SGD achieves:*

$$\frac{1}{T}\sum_{t=1}^{T}\mathbb{E}\|V_k(t)V_k(t)^{\mathsf{T}}\nabla \hat{L}_n(\mathbf{w}_t)\|_2^2 \leq \tilde{O}\left(\frac{k\rho G^2}{n\epsilon}\right) + O\left(\frac{\Lambda G^4\rho^2\gamma_2^2(\mathcal{W},d)\ln p}{m}\right). \tag{3}$$

*Additionally, assuming the principal component of the gradient dominates, i.e., there exist $c > 0$, such that $\frac{1}{T}\sum_{t=1}^{T}\|[\nabla \hat{L}_n(\mathbf{w}_t)]^{\perp}\|_2^2 \leq c\frac{1}{T}\sum_{t=1}^{T}\|[\nabla \hat{L}_n(\mathbf{w}_t)]^{\|}\|_2^2$, we have*

$$\mathbb{E}\|\nabla \hat{L}_n(\mathbf{w}_R)\|_2^2 \leq \tilde{O}\left(\frac{k\rho G^2}{n\epsilon}\right) + O\left(\frac{\Lambda G^4\rho^2\gamma_2^2(\mathcal{W},d)\ln p}{m}\right), \tag{4}$$

*where $\mathbf{w}_R$ is uniformly sampled from $\{\mathbf{w}_1, ..., \mathbf{w}_T\}$.*

Theorem 4 shows that PDP-SGD reduces the error rate of a factor of $p$ to $k$ compared to existing results for non-convex and smooth functions (Wang and Xu, 2019). The error rate also includes a term depending on the $\gamma_2$ function and the eigen-gap $\alpha_t$, i.e., $\Lambda = \sum_{t=1}^{T} 1/\alpha_t^2/T$. This term comes from the subspace reconstruction error. As discussed in the previous section, as the gradients stay in a union of ellipsoids, the $\gamma_2$ is a constant. The term $\Lambda$ depends on the eigen-gap $\alpha_t$, i.e, $\lambda_k(\Sigma_t) - \lambda_{k+1}(\Sigma_t) \geq \alpha_t$. As shown by the Figure 1, along the training trajectory, there are a few dominated eigenvalues and the eigen-gap stays significant (even at the last epoch). Then the term $\Lambda$ will be a constant and the bound scales logarithmically with $p$. If one considers the eigen-gap $\alpha_t$ decays as training proceed, e.g., $\alpha_t = \frac{1}{t^{1/4}}$ for $t > 0$, then we have $\Lambda = O(\sqrt{T})$. In this case, with $T = n^2\epsilon^2$, PDP-SGD requires the public data size $m = O(n\epsilon)$.

## 4 EXPERIMENTS

We empirically evaluate PDP-SGD on training neural networks with two datasets: the MNIST (Le-Cun et al., 1998) and Fashion MNIST (Xiao et al., 2017). We compare the performance of PDP-SGD with the baseline DP-SGD for various privacy levels $\epsilon$. In addition, we also explore a heuristic method, i.e., DP-SGD with random projection by replacing the projector with a $\mathbb{R}^{k\times p}$ Gaussian random projector (Bingham and Mannila, 2001; Blocki et al., 2012). We call this method randomly projected DP-SGD (RPDP-SGD). We present the experimental results after discussing the experimental setup. More details and additional results are in Appendix D.

**Datasets and Network Structure.** The MNIST and Fashion MNIST datasets both consist of 60,000 training examples and 10,000 test examples. To construct the private training set, we randomly sample 10,000 samples from the original training set of MNIST and Fashion MNIST, then we randomly sample 100 samples from the rest to construct the public dataset. Note that the smaller private datasets make the private learning problem more challenging. For both datasets, we use a convolutional neural network that follows the structure in Papernot et al. (2020).

**Training and Hyper-parameter Setting.** Cross-entropy is used as our loss function throughout experiments. The mini-batch size is set to be 250 for both MNIST and Fashion MNIST. For the step size, we follow the grid search method with search space and step size used for DP-SGD, PDP-SGD and RPDP-SGD listed in Appendix D. For training, a fixed budget on the number of epochs i.e., 30 is assigned for the each task. We repeat each experiments 3 times and report the mean and standard deviation of the accuracy on the training and test set. For PDP-SGD, we use Lanczos algorithm to compute the top $k$ eigen-space of the gradient second moment martix on public dataset. We use $k = 50$ for MNIST and $k = 70$ for Fashion MNIST. For RPDP-SGD, we use $k = 800$ for both datasets. Instead of doing the projection for all epochs, we also explored a start point for the projection, i.e., executing the projection from the 1-st epoch, 15-th epoch. We found that for Fashion MNIST, PDP-SGD and RPDP-SGD perform better when starting projection from the 15-th epoch.

**Privacy Parameter Setting:** We consider different choices of the noise scale, i.e., $\sigma = \{18, 14, 10, 8, 6, 4\}$ for MNIST and $\sigma = \{18, 14, 10, 6, 4, 2\}$ for Fashion MNIST. Since gradient

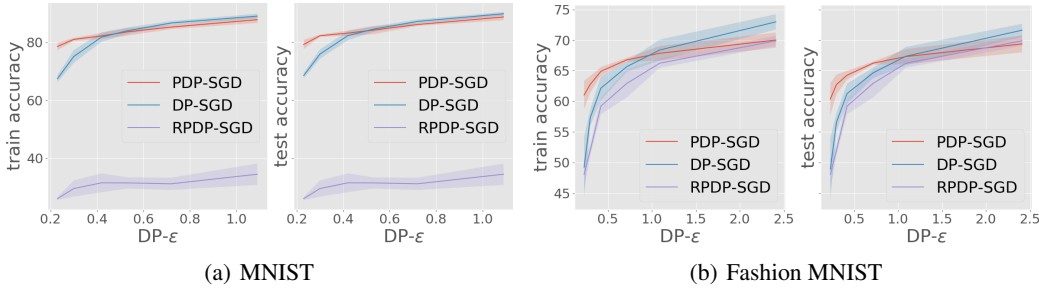

(a) MNIST            (b) Fashion MNIST

Figure 3: Training and test accuracy for DP-SGD, PDP-SGD and RPDP-SGD with different privacy levels for (a) MNIST and (b) Fashion MNIST. The X-axis is the $\epsilon$, and the Y-axis is the train/test accuracy. For small $\epsilon$ regime, which is more favorable for privacy, PDP-SGD outperforms DP-SGD.

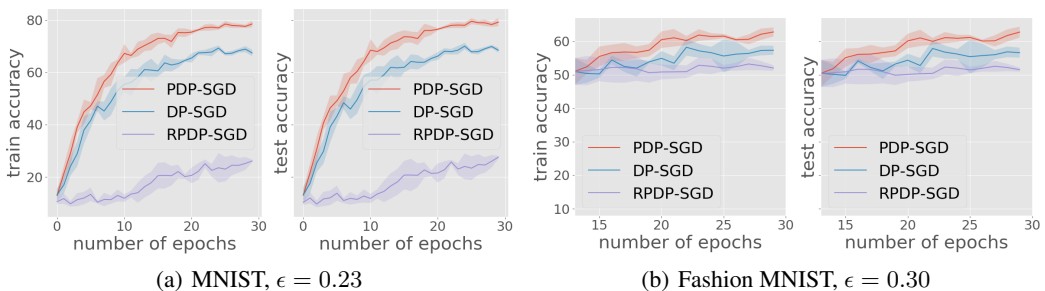

(a) MNIST, $\epsilon = 0.23$           (b) Fashion MNIST, $\epsilon = 0.30$

Figure 4: Training dynamics of DP-SGD, PDP-SGD and RPDP-SGD for (a) MNIST ($\epsilon = 0.23$) and (b) Fashion MNIST ( $\epsilon = 0.30$). The X-axis is the number of epochs, and the Y-axis is the train/test accuracy. For Fashion MNIST, PDP-SGD and RPDP-SGD start projection at 15-th epoch.

norm bound $G$ is unknow for deep learning, we follow the gradient clipping method in Abadi et al. (2016) to guarantee the privacy. We choose gradient clip size to be $1.0$ for both datasets. We follow the Moment Accountant (MA) method (Abadi et al., 2016; Bu et al., 2019) to calculate the accumulated privacy cost, which depends on the number of epochs, the batch size, $\delta$, and noise $\sigma$. With 30 epochs, batch size 250, $10,000$ training samples, and fixing $\delta = 10^{-5}$, the $\epsilon$ is $\{2.41, 1.09, 0.72, 0.42, 0.30, 0.23\}$ for $\sigma \in \{2, 4, 6, 10, 14, 18\}$ for Fashion MNIST. For MNIST, $\epsilon$ is $\{1.09, 0.72, 0.53, 0.42, 0.30, 0.23\}$ for $\sigma \in \{4, 6, 8, 10, 14, 18\}$. Note that $\epsilon$ presented in this paper is w.r.t. a subset i.e., $10,000$ samples from MNIST and Fashion MNIST.

**Experimental Results.** The training accuracy and test accuracy for different $\epsilon$, are reported in Figure 3. For small $\epsilon$ regime, i.e., $\epsilon \leq 0.42$ with MNIST (Figure 3 (a)) and $\epsilon \leq 0.72$ with Fashion MNIST (Figure 3 (b)), PDP-SGD outperforms DP-SGD. For large $\epsilon$ (small noise scale), we think DP-SGD performs better than PDP-SGD because the subspace reconstruction error dominates the error from the injected noise. For most choices of $\epsilon$, RPDP-SGD fails to improve the accuracy over DP-SGD because the subspace reconstruction error introduced by the random projector is larger than the noise error reduced by projection. To the best of our knowledge, we noticed that when $\epsilon < 1$ for MNIST, PDP-SGD and DP-SGD perform better than the benchmark reported in Papernot et al. (2020) even with a subset from MNIST (Figure 3 (a)). We acknowledge that Papernot et al. (2020) report the test accuracy as a training dynamic in terms of privacy loss $\epsilon$. Figure 4 provides two examples of the training dynamics, i.e., MNIST with $\epsilon = 0.23$ and Fashion MNIST with $\epsilon = 0.30$, showing that PDP-SGD outperforms DP-SGD for large noise scale since PDP-SGD efficiently reduces the noise. We also validate this observation on larger training samples in Appendix D.

We also study the role of projection dimension $k$ and pubic sample size $m$. Figure 5(a) and Figure 5(b) present the training and test accuracy for PDP-SGD with $k \in \{10, 20, 30, 50\}$ and PDP-SGD with $m \in \{50, 100, 150\}$ for $\epsilon = 0.23$ for MNIST dataset. Among the choices of $k$, PDP-SGD with $k = 50$ achieves the best accuracy. DP-SGD with $k = 10$ proceeds slower than the rest, due to the larger reconstruction error introduced by projecting the gradient to a much smaller subspace, i..e, $k = 10$. However, compared to the gradient dimension $p \approx 25,000$, it is impressive that PDP-SGD with $k = 50$ can achieve better accuracy than DP-SGD for a certain range of $\epsilon$. Figure 5(b) shows

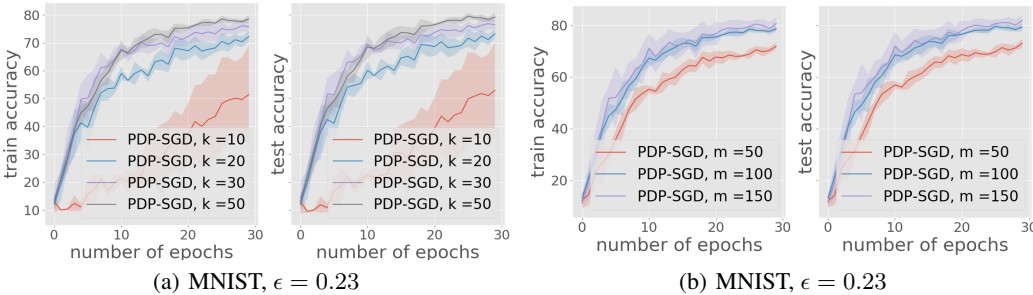

(a) MNIST, $\epsilon = 0.23$           (b) MNIST, $\epsilon = 0.23$

Figure 5: Training accuracy and test accuracy for (a) PDP-SGD with $k = \{10, 20, 30, 50\}$; (b) PDP-SGD with $m = \{50, 100, 150\}$ for MNIST ( $\epsilon = 0.23$). The X-axis and Y-axis refer to Figure 4. The performance of PDP-SGD increases as projection dimension $k$ and public sample size $m$ increase.

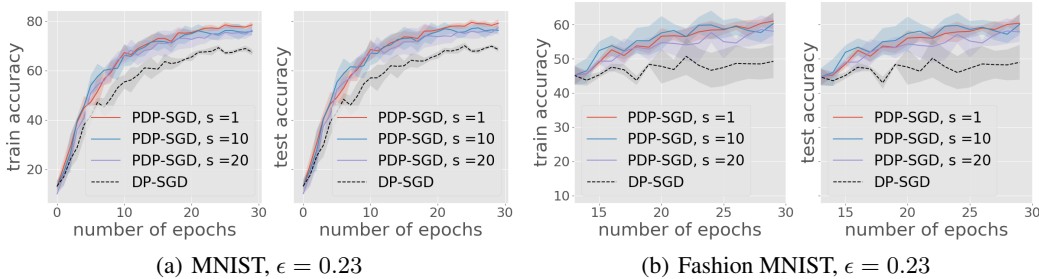

(a) MNIST, $\epsilon = 0.23$           (b) Fashion MNIST, $\epsilon = 0.23$

Figure 6: Training and test accuracy for PDP-SGD with different frequency of eigen-space computation for (a) MNIST ($\epsilon = 0.23$) and (b) Fashion MNIST ($\epsilon = 0.23$). $s = \{1, 10, 20\}$ is the frequency of subspace update, i.e., compute the eigen-space every $s$ iterates. The X-axis and Y-axis refer to Figure 4. For Fashion MNIST, PDP-SGD starts projection at 15-th epoch. PDP-SGD with a reduced eigen-space computation also improves the accuracy over DP-SGD.

that the accuracy of PDP-SGD improves as the $m$ increases from 50 to 150. This is consistent with the theoretical analysis that increasing $m$ helps to reduce the subspace reconstruction error. Also, PDP-SGD with $m = 100$ performs similar to PDP-SGD with $m = 150$. The results suggest that while a small number of public datasets are not sufficient for training an accurate predictor, they provide useful gradient subspace projection and accuracy improvement over DP-SGD.

To reduce the computation complexity introduced by eigen-value decomposition, we explored PDP-SGD with sparse eigen-space computation, i.e., update the projector every $s$ iterates. Note that PDP-SGD with $s = 1$ means computing the top eigen-space at every iteration. Figure 6 reports PDP-SGD with $s = \{1, 10, 20\}$ for (a) MNIST and (b) Fashion MNIST showing that PDP-SGD with a reduced eigen-space computation also outperforms DP-SGD, even though there is a mild decay for PDP-SGD with fewer eigen-space computations.

## 5 CONCLUSION

While DP-SGD and variants have been well studied for private ERM, the error rate of DP-SGD depends on the ambient dimension $p$. In this paper, we aim to bypass such dependence by leveraging the low-dimensional structure of the observed gradients in the training of deep networks. We propose PDP-SGD which projects the noisy gradient to an approximated subspace evaluated on a public dataset. We show theoretically that PDP-SGD can obtain (near) dimension-independent error rate. We evaluate the proposed algorithms on two popular deep learning tasks and demonstrate the empirical advantages of PDP-SGD.

## ACKNOWLEDGEMENT

The research was supported by NSF grants IIS-1908104, OAC-1934634, IIS-1563950, a Google Faculty Research Award, a J.P. Morgan Faculty Award, and a Mozilla research grant. We would like to thank the Minnesota Super-computing Institute (MSI) for providing computational resources and support.

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

## A  Uniform Convergence for Subspaces: Proofs for Section 3.1

In this section, we provide the proofs for Section 3.1. We first show that the second moment matrix $M_t$ converges to the population second moment matrix $\Sigma_t$ uniform over all iterations $t \in [T]$, i.e., $\sup_{t \in [T]} \|M_t - \Sigma_t\|$. Then we show that the top-$k$ subspace of $M_t$ uniformly converges to the top-$k$ subspace of $\Sigma_t$, i.e., $\|\hat{V}_k(t)\hat{V}_k(t)^\mathsf{T} - V_k(t)V_k(t)^\mathsf{T}\|$ for all $t \in [T]$. Our bound depends on $\gamma_2(\mathcal{W}, d)$ where $\mathcal{W}$ is the set of all possible parameters along the training trajectory. In Section A.2, we show that the bound can be derived by $\gamma_2(\mathcal{M}, d)$ as well, where $\mathcal{M}$ is the set of population gradients along the training trajectory. Then, we provide examples of the set $\mathcal{M}$ and corresponding value of $\gamma_2(\mathcal{M}, d)$.

### A.1  Uniform Convergence Bound

Our proofs of Theorem 2 heavily rely on the advanced probability tool, Generic Chaining (GC) (Talagrand, 2014). Typically the results in generic chaining are characterized by the so-called $\gamma_2$ function (see Definition 2). Talagrand (2014) shows that for a process $(X_t)_{t \in T}$ and a given metric space $(T, d)$, if $(X_t)_{t \in T}$ satisfies the increment condition

$$\forall u > 0, \mathbb{P}\left(|X_s - X_t| \geq u\right) \leq 2 \exp\left(-\frac{u^2}{2d(s,t)^2}\right), \tag{5}$$

then the size of the process can be bounded as

$$\mathbb{E} \sup_{t \in T} X_t \leq c\gamma_2(T, d), \tag{6}$$

with $c$ to be an absolute constant.

To apply the GC result to establish the Theorem 2, we treat $\|M_t - \Sigma_t\|_2$ as the process $X_t$ over the iterations. In detail, since $M_t = \frac{1}{m}\sum_{i=1}^m \nabla\ell(\mathbf{w}_t, \tilde{z}_i)\nabla\ell(\mathbf{w}_t, \tilde{z}_i)^\mathsf{T}$, and $\Sigma_t = \mathbb{E}_{z \sim \mathcal{P}}[\nabla\ell(\mathbf{w}_t, z)\nabla\ell(\mathbf{w}_t, z)^\mathsf{T}]$, the $\|M_t - \Sigma_t\|_2$ is a random process indexed by $\mathbf{w}_t \in \mathcal{W}$, with $\mathcal{W}$ to be the set of all possible iterates obtained by the algorithm.

We first show that the variable $\|M_t - \Sigma_t\|_2$ satisfies the increment condition as stated in equation 5 in Lemma 1. Before we present the proof of Lemma 1, we introduce the Ahlswede-Winter Inequality (Horn and Johnson, 2012; Wainwright, 2019), which will be used in the proof of Lemma 1. Ahlswede-Winter Inequality shows that positive semi-definite random matrix with bounded spectral norm concentrates to its expectation with high probability.

**Theorem 5** *(Ahlswede-Winter Inequality) Let $Y$ be a random, symmetric, positive semi-definite $p \times p$ matrix. such that such that $\|\mathbb{E}[Y]\| \leq 1$. Suppose $\|Y\| \leq R$ for some fixed scalar $R \geq 1$. Let $\{Y_1, \ldots, Y_m\}$ be independent copies of $Y$ (i.e., independently sampled matrices with the same distribution as $Y$). For any $u \in [0, 1]$, we have*

$$\mathbb{P}\left(\left\|\frac{1}{m}\sum_{i=1}^m Y_i - \mathbb{E}[Y_i]\right\|_2 > u\right) \leq 2p \cdot \exp\left(-mu^2/4R\right). \tag{7}$$

To make the argument clear, we use a more informative notation for $M_t$ and $\Sigma_t$. Recall the notation of $M_t$ and $\Sigma_t$ such that

$$M_t = \frac{1}{m}\sum_{i=1}^m \nabla\ell(\mathbf{w}_t, \tilde{z}_i)\nabla\ell(\mathbf{w}_t, \tilde{z}_i)^\mathsf{T}, \tag{8}$$

and

$$\Sigma_t = \mathbb{E}_{z \sim \mathcal{P}}[\nabla\ell(\mathbf{w}_t, z)\nabla\ell(\mathbf{w}_t, z)^\mathsf{T}], \tag{9}$$

given the dataset $S_h = \{\tilde{z}_1, .., \tilde{z}_m\}$ and distribution $\mathcal{P}$ where $\tilde{z}_i \sim \mathcal{P}$ for $i \in [m]$, the $M_t$ and $\Sigma_t$ are functions of parameter $\mathbf{w}_t$, so we use $M(\mathbf{w}_t)$ and $\Sigma(\mathbf{w}_t)$ for $M_t$ and $\Sigma_t$ interchangeably in the rest of this section, i..e,

$$M(\mathbf{w}) = \frac{1}{m}\sum_{i=1}^m \nabla\ell(\mathbf{w}, \tilde{z}_i)\nabla\ell(\mathbf{w}, \tilde{z}_i)^\mathsf{T} \tag{10}$$

and

$$\Sigma(\mathbf{w}) = \mathbb{E}_{z \sim \mathcal{P}} \left[ \nabla \ell (\mathbf{w}, z) \nabla \ell (\mathbf{w}, z)^{\mathsf{T}} \right]. \tag{11}$$

**Lemma 1** *With Assumption 2 and 1 hold, for any* $\mathbf{w}, \mathbf{w}' \in \mathcal{W}$ *and* $\forall u > 0$*, we have*

$$\mathbb{P} \left( \|M(\mathbf{w}) - \Sigma(\mathbf{w})\|_2 - \|M(\mathbf{w}') - \Sigma(\mathbf{w}')\|_2 \geq \frac{u}{\sqrt{m}} \cdot 4G\rho d(\mathbf{w}, \mathbf{w}') \right) \leq 2p \cdot \exp \left( -u^2/4 \right), \tag{12}$$

*where* $d : \mathcal{W} \times \mathcal{W} \mapsto \mathbb{R}$ *is the pseudo-metric in Assumption 2.*

*Proof:*  We consider random variable

$$X_i = \nabla \ell(\mathbf{w}, \tilde{z}_i) \nabla \ell(\mathbf{w}, \tilde{z}_i)^{\mathsf{T}} - \nabla \ell(\mathbf{w}', \tilde{z}_i) \nabla \ell(\mathbf{w}', \tilde{z}_i)^{\mathsf{T}} + 2G\rho d(\mathbf{w}, \mathbf{w}') \mathbb{I}_p, \tag{13}$$

where $\mathbb{I}_p \in \mathbb{R}^{p \times p}$ is the identity matrix.

Note that $2G\rho d(\mathbf{w}, \mathbf{w}')$ is deterministic and the randomness of $X_i$ comes from $\nabla \ell(\mathbf{w}, \tilde{z}_i)$ and $\nabla \ell(\mathbf{w}', \tilde{z}_i)$.

By triangle inequality and the construction of $X_i$, we have

$$\|M(\mathbf{w}) - \Sigma(\mathbf{w})\|_2 - \|M(\mathbf{w}') - \Sigma(\mathbf{w}')\|_2$$
$$= \|M(\mathbf{w}) - \mathbb{E}[M(\mathbf{w})]\|_2 - \|M(\mathbf{w}') - \mathbb{E}[M(\mathbf{w}')]\|_2$$
$$\leq \|M(\mathbf{w}) - \mathbb{E}[M(\mathbf{w})] - (M(\mathbf{w}') - \mathbb{E}[M(\mathbf{w}')])\|_2$$
$$= \|M(\mathbf{w}) - M(\mathbf{w}') - \mathbb{E}[(M(\mathbf{w}) - M(\mathbf{w}'))]\|_2$$
$$= \left\| \frac{1}{m} \sum_{i=1}^{m} (\nabla \ell(\mathbf{w}, \tilde{z}_i) \nabla \ell(\mathbf{w}, \tilde{z}_i)^{\mathsf{T}} - \nabla \ell(\mathbf{w}', \tilde{z}_i) \nabla \ell(\mathbf{w}', \tilde{z}_i)^{\mathsf{T}}) - \mathbb{E}\left[ \nabla \ell(\mathbf{w}, \tilde{z}_i) \nabla \ell(\mathbf{w}, \tilde{z}_i)^{\mathsf{T}} - \nabla \ell(\mathbf{w}', \tilde{z}_i) \nabla \ell(\mathbf{w}', \tilde{z}_i)^{\mathsf{T}} \right] \right\|_2$$
$$= \left\| \frac{1}{m} \sum_{i=1}^{m} (\nabla \ell(\mathbf{w}, \tilde{z}_i) \nabla \ell(\mathbf{w}, \tilde{z}_i)^{\mathsf{T}} - \nabla \ell(\mathbf{w}', \tilde{z}_i) \nabla \ell(\mathbf{w}', \tilde{z}_i)^{\mathsf{T}} + 2G\rho d(\mathbf{w}, \mathbf{w}') \mathbb{I}_p) \right.$$
$$\left. - \mathbb{E}\left[ \nabla \ell(\mathbf{w}, \tilde{z}_i) \nabla \ell(\mathbf{w}, \tilde{z}_i)^{\mathsf{T}} - \nabla \ell(\mathbf{w}', \tilde{z}_i) \nabla \ell(\mathbf{w}', \tilde{z}_i)^{\mathsf{T}} + 2G\rho d(\mathbf{w}, \mathbf{w}') \mathbb{I}_p \right] \right\|_2$$
$$= \left\| \frac{1}{m} \sum X_i - \mathbb{E}[X_i] \right\|_2 \tag{14}$$

To apply Theorem 5 for $\left\| \frac{1}{m} \sum X_i - \mathbb{E}[X_i] \right\|_2$, we first show that the random symmetric matrix $X_i$ is positive semi-definite.

By Assumption 1 and Assumption 2 and definition

$$\|\nabla \ell(\mathbf{w}, \tilde{z}_i) \nabla \ell(\mathbf{w}, \tilde{z}_i)^{\mathsf{T}} - \nabla \ell(\mathbf{w}', \tilde{z}_i) \nabla \ell(\mathbf{w}', \tilde{z}_i)^{\mathsf{T}}\|_2$$
$$= \sup_{\mathbf{x} : \|\mathbf{x}\| = 1} \mathbf{x}^{\mathsf{T}} \left( \nabla \ell(\mathbf{w}, \tilde{z}_i) \nabla \ell(\mathbf{w}, \tilde{z}_i)^{\mathsf{T}} - \nabla \ell(\mathbf{w}', \tilde{z}_i) \nabla \ell(\mathbf{w}', \tilde{z}_i)^{\mathsf{T}} \right) \mathbf{x}$$
$$= \sup_{\mathbf{x} : \|\mathbf{x}\| = 1} \mathbf{x}^{\mathsf{T}} \nabla \ell(\mathbf{w}, \tilde{z}_i) \nabla \ell(\mathbf{w}, \tilde{z}_i)^{\mathsf{T}} \mathbf{x} - \mathbf{x}^{\mathsf{T}} \nabla \ell(\mathbf{w}', \tilde{z}_i) \nabla \ell(\mathbf{w}', \tilde{z}_i)^{\mathsf{T}} \mathbf{x}$$
$$= \sup_{\mathbf{x} : \|\mathbf{x}\| = 1} \langle \mathbf{x}, \nabla \ell(\mathbf{w}, \tilde{z}_i) \rangle^2 - \langle \mathbf{x}, \nabla \ell(\mathbf{w}', \tilde{z}_i) \rangle^2$$
$$= \sup_{\mathbf{x} : \|\mathbf{x}\| = 1} (\langle \mathbf{x}, \nabla \ell(\mathbf{w}, \tilde{z}_i) \rangle + \langle \mathbf{x}, \nabla \ell(\mathbf{w}', \tilde{z}_i) \rangle) (\langle \mathbf{x}, \nabla \ell(\mathbf{w}, \tilde{z}_i) \rangle - \langle \mathbf{x}, \nabla \ell(\mathbf{w}', \tilde{z}_i) \rangle)$$
$$\leq \sup_{\mathbf{x} : \|\mathbf{x}\| = 1} 2G (\langle \mathbf{x}, \nabla \ell(\mathbf{w}, \tilde{z}_i) \rangle - \langle \mathbf{x}, \nabla \ell(\mathbf{w}', \tilde{z}_i) \rangle)$$
$$= 2G\|\nabla \ell(\mathbf{w}, \tilde{z}_i) - \nabla \ell(\mathbf{w}', \tilde{z}_i)\|_2$$
$$\leq 2G\rho d(\mathbf{w}, \mathbf{w}') \tag{15}$$

For any non-zero vector $\mathbf{x} \in \mathbb{R}^p$, we have

$$
\begin{aligned}
\mathbf{x}^\intercal X_i \mathbf{x} &= \mathbf{x}^\intercal \left( \nabla\ell(\mathbf{w},\tilde{z}_i)\nabla\ell(\mathbf{w},\tilde{z}_i)^\intercal - \nabla\ell(\mathbf{w}',\tilde{z}_i)\nabla\ell(\mathbf{w}',\tilde{z}_i)^\intercal + 2G\rho d(\mathbf{w},\mathbf{w}')\mathbb{I}_p \right) \mathbf{x} \\
&= \mathbf{x}^\intercal \left( \nabla\ell(\mathbf{w},\tilde{z}_i)\nabla\ell(\mathbf{w},\tilde{z}_i)^\intercal - \nabla\ell(\mathbf{w}',\tilde{z}_i)\nabla\ell(\mathbf{w}',\tilde{z}_i)^\intercal \right) \mathbf{x} + 2G\rho d(\mathbf{w},\mathbf{w}')\|\mathbf{x}\|_2^2 \\
&\geq -\left\| \nabla\ell(\mathbf{w},\tilde{z}_i)\nabla\ell(\mathbf{w},\tilde{z}_i)^\intercal - \nabla\ell(\mathbf{w}',\tilde{z}_i)\nabla\ell(\mathbf{w}',\tilde{z}_i)^\intercal \right\|_2 \|\mathbf{x}\|_2^2 + 2G\rho d(\mathbf{w},\mathbf{w}')\|\mathbf{x}\|_2^2 \\
&= \left( 2G\rho d(\mathbf{w},\mathbf{w}') - \left\| \nabla\ell(\mathbf{w},\tilde{z}_i)\nabla\ell(\mathbf{w},\tilde{z}_i)^\intercal - \nabla\ell(\mathbf{w}',\tilde{z}_i)\nabla\ell(\mathbf{w}',\tilde{z}_i)^\intercal \right\|_2 \right) \|\mathbf{x}\|_2^2 \\
&\overset{(a)}{\geq} 0,
\end{aligned}
\tag{16}
$$

where (a) is true because $\left\| \nabla\ell(\mathbf{w},\tilde{z}_i)\nabla\ell(\mathbf{w},\tilde{z}_i)^\intercal - \nabla\ell(\mathbf{w}',\tilde{z}_i)\nabla\ell(\mathbf{w}',\tilde{z}_i)^\intercal \right\|_2 \leq 2G\rho d(\mathbf{w},\mathbf{w}')$ as shown in equation 15.

Let $Y_i = \frac{X_i}{4G\rho d(\mathbf{w},\mathbf{w}')}$, with equation 15, we have

$$
\begin{aligned}
\|Y_i\|_2 &= \frac{\left\| \nabla\ell(\mathbf{w},\tilde{z}_i)\nabla\ell(\mathbf{w},\tilde{z}_i)^\intercal - \nabla\ell(\mathbf{w}',\tilde{z}_i)\nabla\ell(\mathbf{w}',\tilde{z}_i)^\intercal + 2G\rho d(\mathbf{w},\mathbf{w}')\mathbb{I}_p \right\|_2}{4G\rho d(\mathbf{w},\mathbf{w}')} \\
&\leq \frac{\left\| \nabla\ell(\mathbf{w},\tilde{z}_i)\nabla\ell(\mathbf{w},\tilde{z}_i)^\intercal - \nabla\ell(\mathbf{w}',\tilde{z}_i)\nabla\ell(\mathbf{w}',\tilde{z}_i)^\intercal \right\|_2 + \left\| 2G\rho d(\mathbf{w},\mathbf{w}')\mathbb{I}_p \right\|_2}{4G\rho d(\mathbf{w},\mathbf{w}')} \\
&\leq \frac{2G\rho d(\mathbf{w},\mathbf{w}') + 2G\rho d(\mathbf{w},\mathbf{w}')}{4G\rho d(\mathbf{w},\mathbf{w}')} \\
&= 1.
\end{aligned}
\tag{17}
$$

So that $\|Y_i\|_2 \leq 1$ and $\|\mathbb{E}[Y_i]\|_2 \leq 1$. Then, from Theorem 5, with $R = 1$, we have for any $u \in [0,1]$

$$
\mathbb{P}\left( \left\| \frac{1}{m}\sum_{i=1}^m Y_i - \mathrm{E}[Y_i] \right\| > u \right) \leq 2p \cdot \exp\left( -mu^2/4 \right).
\tag{18}
$$

Note that $\left\| \frac{1}{m}\sum_{i=1}^m Y_i - \mathrm{E}[Y_i] \right\|$ is always bounded by 1 since $\|Y_i\|_2 \leq 1$ and $\|\mathbb{E}[Y_i]\|_2 \leq 1$. So the above inequality holds for any $u > 1$ with probability 0 which is bounded by $2p \cdot \exp\left( -mu^2/4 \right)$. So that we have for any $u > 0$,

$$
\mathbb{P}\left( \left\| \frac{1}{m}\sum_{i=1}^m Y_i - \mathrm{E}[Y_i] \right\| > u \right) \leq 2p \cdot \exp\left( -mu^2/4 \right).
\tag{19}
$$

So that for any $u > 0$,

$$
\mathbb{P}\left( \left\| \frac{1}{m}\sum_{i=1}^m X_i - \mathrm{E}[X_i] \right\| > \frac{u}{\sqrt{m}} \cdot 4G\rho d(\mathbf{w},\mathbf{w}') \right) \leq 2p \cdot \exp\left( -u^2/4 \right).
\tag{20}
$$

Combine equation 20 and equation 14, we have

$$
\begin{aligned}
&\mathbb{P}\left( \|M(\mathbf{w}) - \Sigma(\mathbf{w})\|_2 - \|M(\mathbf{w}') - \Sigma(\mathbf{w}')\|_2 \geq \frac{u}{\sqrt{m}} \cdot 4G\rho d(\mathbf{w},\mathbf{w}') \right) \\
&= \mathbb{P}\left( \|M(\mathbf{w}) - \mathbb{E}[M(\mathbf{w})]\|_2 - \|M(\mathbf{w}') - \mathbb{E}[M(\mathbf{w}')]\|_2 > \frac{u}{\sqrt{m}} \cdot 4G\rho d(\mathbf{w},\mathbf{w}') \right) \\
&\leq \mathbb{P}\left( \left\| \frac{1}{m}\sum_{i=1}^m X_i - \mathrm{E}[X_i] \right\| > \frac{u}{\sqrt{m}} \cdot 4G\rho d(\mathbf{w},\mathbf{w}') \right) \\
&\leq 2p \cdot \exp\left( -u^2/4 \right)
\end{aligned}
\tag{21}
$$

That completes the proof. ∎

Based on the above result, now we come to the proof of Theorem 2. The proof follows the Generic Chaining argument, i.e., Chapter 2 of Talagrand (2014).

**Theorem 2 (Second Moment Concentration)** *Under Assumption 1, 2, the second moment matrix of the public gradient $M_t = \frac{1}{m}\sum_{i=1}^m \nabla\ell(\mathbf{w}_t, \tilde{z}_i)\nabla\ell(\mathbf{w}_t, \tilde{z}_i)^\intercal$ approximates the population second moment matrix $\Sigma_t = \mathbb{E}_{z\sim\mathcal{P}}[\nabla\ell(\mathbf{w}_t, z)\nabla\ell(\mathbf{w}_t, z)^\intercal]$ uniformly over all iterations, i.e., for any $u > 0$,*

$$\sup_{t\in[T]} \|M_t - \Sigma_t\|_2 \leq O\left(\frac{uG\rho\sqrt{\ln p}\gamma_2(\mathcal{W}, d)}{\sqrt{m}}\right), \tag{1}$$

*with probability at least $1 - c\exp\left(-u^2/4\right)$, where $c$ is an absolute constant.*

*Proof:* Note that equation equation 1 is a uniform bound over iteration $t \in [T]$. To bound $\sup_{t\in[T]} \|M_t - \Sigma_t\|_2$, it is sufficient to bound

$$\sup_{\mathbf{w}\in\mathcal{W}} \|M(\mathbf{w}) - \Sigma(\mathbf{w})\|_2, \tag{22}$$

where $\mathcal{W}$ contains all the possible trajectorys of $\mathbf{w}_1, ..., \mathbf{w}_T$.

We consider a sequence of subsets $\mathcal{W}_n$ of $\mathcal{W}$, and

$$\operatorname{card}\mathcal{W}_n \leq N_n, \tag{23}$$

where $N_0 = 1$; $N_n = 2^{2^n}$ if $n \geq 1$.

Let $\pi_n(\mathbf{w}) \in \mathcal{W}_n$ be the approximation of any $\mathbf{w} \in \mathcal{W}$. We decompose the $\|M(\mathbf{w}) - \Sigma(\mathbf{w})\|_2$ as

$$\|M(\mathbf{w}) - \Sigma(\mathbf{w})\|_2 - \|M(\pi_0(\mathbf{w})) - \Sigma(\pi_0(\mathbf{w}))\|_2$$
$$= \sum_{n\geq 1}\left(\|M(\pi_n(\mathbf{w})) - \Sigma(\pi_n(\mathbf{w}))\|_2 - \|M(\pi_{n-1}(\mathbf{w})) - \Sigma(\pi_{n-1}(\mathbf{w}))\|_2\right), \tag{24}$$

which holds since $\pi_n(\mathbf{w}) = \mathbf{w}$ for $n$ large enough.

Based on Lemma 1, for any $u > 0$, we have

$$\|M(\pi_n(\mathbf{w})) - \Sigma(\pi_n(\mathbf{w}))\|_2 - \|M(\pi_{n-1}(\mathbf{w})) - \Sigma(\pi_{n-1}(\mathbf{w}))\|_2 \geq \frac{u}{\sqrt{m}}\cdot 4G\rho d(\pi_n(\mathbf{w}), \pi_{n-1}(\mathbf{w})), \tag{25}$$

with probability at most $2p\exp(-\frac{u^2}{4})$.

For any $n > 0$ and $\mathbf{w} \in \mathcal{W}$, the number of possible pairs $(\pi_n(\mathbf{w}), \pi_{n-1}(\mathbf{w}))$ is

$$\operatorname{card}\mathcal{W}_n \cdot \operatorname{card}\mathcal{W}_{n-1} \leq N_n N_{n-1} \leq N_{n+1} = 2^{2^{n+1}}. \tag{26}$$

Apply union bound over all the possible pairs of $(\pi_n(\mathbf{w}), \pi_{n-1}(\mathbf{w}))$, following Talagrand (2014) (Chapter 2.2), for any $n > 0$, $u > 0$, and $\mathbf{w} \in \mathcal{W}$, we have

$$\|M(\pi_n(\mathbf{w})) - \Sigma(\pi_n(\mathbf{w}))\|_2 - \|M(\pi_{n-1}(\mathbf{w})) - \Sigma(\pi_{n-1}(\mathbf{w}))\|_2 \geq u2^{n/2}d(\pi_n(\mathbf{w}), \pi_{n-1}(\mathbf{w}))\cdot\frac{4G\rho}{\sqrt{m}} \tag{27}$$

with probability

$$\sum_{n\geq 1} 2p\cdot 2^{2^{n+1}}\exp\left(-u^2 2^{n-2}\right) \leq c'p\exp\left(-\frac{u^2}{4}\right), \tag{28}$$

where $c'$ is a universal constant.

Then we have

$$\sum_{n\geq 1}\left(\|M(\pi_n(\mathbf{w})) - \Sigma(\pi_n(\mathbf{w}))\|_2 - \|M(\pi_{n-1}(\mathbf{w})) - \Sigma(\pi_{n-1}(\mathbf{w}))\|_2\right)$$

$$\geq \sum_{n\geq 1} u2^{n/2}d(\pi_n(\mathbf{w}), \pi_{n-1}(\mathbf{w}))\cdot\frac{4G\rho}{\sqrt{m}}$$

$$\geq \sum_{n\geq 0} u2^{n/2}d(\mathbf{w}, \mathcal{W}_n)\cdot\frac{4G\rho}{\sqrt{m}} \tag{29}$$

with probability at most $c'p \exp\left(-\frac{u^2}{4}\right)$.

From Theorem 5, let $Y_i = \frac{\nabla \ell(\pi_0(\mathbf{w}), \tilde{z}_i) \nabla \ell(\pi_0(\mathbf{w}), \tilde{z}_i)^\intercal}{G}$, so that $\|Y_i\| \leq 1$ and $\|\mathbb{E}[Y_i]\| \leq 1$. Then we have

$$\|M(\pi_0(\mathbf{w})) - \Sigma(\pi_0(\mathbf{w}))\|_2 \geq u \frac{G}{\sqrt{m}}, \tag{30}$$

with ptobability at most $2p \exp\left(-\frac{u^2}{4}\right)$.

Combine equation 24, equation 29 and equation 30, we have

$$\sup_{\mathbf{w} \in \mathcal{W}} \|M(\mathbf{w}) - \Sigma(\mathbf{w})\|_2 \geq \sup_{\mathbf{w} \in \mathcal{W}} \sum_{n \geq 0} u 2^{n/2} d(\mathbf{w}, \mathcal{W}_n) \cdot \frac{4G\rho}{\sqrt{m}} + u \frac{G}{\sqrt{m}}$$

$$= u \left( \frac{G\left(4\rho\gamma_2(\mathcal{W}, d) + 1\right)}{\sqrt{m}} \right) \tag{31}$$

with probability at most $(c' + 2)p \exp\left(-\frac{u^2}{4}\right)$.

That completes the proof. ∎

Now we provide the proof of Theorem 3.

**Theorem 3 (Subspace Closeness)** *Under Assumption 1 and 2, with $V_k(t)$ to be the top-$k$ eigenvectors of the population second moment matrix $\Sigma_t$ and $\alpha_t$ be the eigen-gap at $t$-th iterate such that $\lambda_k(\Sigma_t) - \lambda_{k+1}(\Sigma_t) \geq \alpha_t$, for the $\hat{V}_k(t)$ in Algorithm 1, if $m \geq \frac{O\left(G\rho\sqrt{\ln p}\gamma_2(\mathcal{W}, d)\right)^2}{\min_t \alpha_t^2}$, we have*

$$\mathbb{E}\left[\|\hat{V}_k(t)\hat{V}_k(t)^\intercal - V_k(t)V_k(t)^\intercal\|_2\right] \leq O\left(\frac{G\rho\sqrt{\ln p}\gamma_2(\mathcal{W}, d)}{\alpha_t\sqrt{m}}\right), \forall t \in [T]. \tag{2}$$

*Proof:* Recall that $\hat{V}_k(t)$ is the top-$k$ eigenspace of $M_t$. Let $V_k(t)$ be the top-$k$ eigenspace of $\Sigma_t$.

$$\hat{V}_k(t)\hat{V}_k(t)^\intercal - V_k(t)V_k(t)^\intercal = \Pi_{M_t}^{(k)}(\mathbb{I} - \Pi_{\Sigma_t}^{(k)}) + (\mathbb{I} - \Pi_{M_t}^{(k)})\Pi_{\Sigma_t}^{(k)} \tag{32}$$

$$\Rightarrow \quad \|\hat{V}_k(t)\hat{V}_k(t)^\intercal - V_k(t)V_k(t)^\intercal\|_2 \leq \|\Pi_{M_t}^{(k)}(\mathbb{I} - \Pi_{\Sigma_t}^{(k)})\|_2 + \|(\mathbb{I} - \Pi_{M_t}^{(k)})\Pi_{\Sigma_t}^{(k)}\|_2. \tag{33}$$

where $\Pi_{M_t}^{(k)} = \hat{V}_k(t)\hat{V}_k(t)^\intercal$ denotes the projection to the top-$k$ subspace of the symmetric PSD $M_t$ and $\Pi_{\Sigma_t}^{(k)} = V_k(t)V_k(t)^\intercal$ denotes the projection to the top-$k$ subspace of the symmetric PSD $\Sigma_t$. Then, from Davis-Kahan (Corollary 8 in McSherry (2004)) and using the fact for symmetric PSD matrices eigen-values and singular values are the same, we have

$$\|\Pi_{M_t}^{(k)}(\mathbb{I} - \Pi_{\Sigma_t}^{(k)})\|_2 \leq \frac{\|M_t - \Sigma_t\|_2}{\lambda_k(M_t) - \lambda_{k+1}(\Sigma_t)} \tag{34}$$

$$\|(\mathbb{I} - \Pi_{M_t}^{(k)})\Pi_{\Sigma_t}^{(k)}\|_2 \leq \frac{\|M_t - \Sigma_t\|_2}{\lambda_k(\Sigma_t) - \lambda_{k+1}(M_t)}. \tag{35}$$

Recall, from Horn and Johnson (2012) (Section 4.3) and Golub and Van Loan (1996) (Section 8.1.2), e.g., Corollary 8.1.6, we have

$$|\lambda_k(M_t) - \lambda_k(\Sigma_t)| \leq \|M_t - \Sigma_t\|_2. \tag{36}$$

From Theorem 2 and Lemma 2, with $Y = \|M_t - \Sigma_t\|_2$, $A = c$, and $B = \frac{4G\rho\sqrt{\ln p}\gamma_2(\mathcal{W}, d)}{\sqrt{m}}$, we have

$$\mathbb{E}\left[\|M_t - \Sigma_t\|_2\right] \leq O\left(\frac{G\rho\sqrt{\ln p}\gamma_2(\mathcal{W}, d)}{\sqrt{m}}\right). \tag{37}$$

Let $c_0 = O\left(G\rho\sqrt{\ln p}\gamma_2(\mathcal{W}, d)\right)$. For $m \geq \frac{c_0^2}{\alpha_t^2}$, we have

$$\mathbb{E}\left[\|M_t - \Sigma_t\|_2\right] \leq \frac{c_0}{\sqrt{m}} \leq \frac{\alpha_t}{2}. \tag{38}$$

Then, for equation 34, we have

$$
\begin{aligned}
\|\Pi_{M_t}^{(k)}(\mathbb{I} - \Pi_{\Sigma_t}^{(k)})\|_2 &\leq \frac{\|M_t - \Sigma_t\|_2}{\lambda_k(M_t) - \lambda_{k+1}(\Sigma_t)} \\
&= \frac{\|M_t - \Sigma_t\|_2}{(\lambda_k(\Sigma_t) - \lambda_{k+1}(\Sigma_t)) - (\lambda_k(\Sigma_t) - \lambda_k(M_t))} \\
&\leq \frac{\|M_t - \Sigma_t\|_2}{\alpha_t - \|M_t - \Sigma_t\|_2}.
\end{aligned}
\tag{39}
$$

Then, for equation 35, we have

$$
\begin{aligned}
\|(\mathbb{I} - \Pi_{M_t}^{(k)})\Pi_{\Sigma_t}^{(k)}\|_2 &\leq \frac{\|M_t - \Sigma_t\|_2}{\lambda_k(\Sigma_t) - \lambda_{k+1}(M_t)} \\
&= \frac{\|M_t - \Sigma_t\|_2}{(\lambda_k(\Sigma_t) - \lambda_{k+1}(\Sigma_t)) + (\lambda_{k+1}(\Sigma_t) - \lambda_{k+1}(M_t))} \\
&\leq \frac{\|M_t - \Sigma_t\|_2}{\alpha_t - \|M_t - \Sigma_t\|_2}.
\end{aligned}
\tag{40}
$$

Combining these two bounds and equation 37, with $c_0 = O\left(G\rho\sqrt{\ln p}\gamma_2(\mathcal{W}, d)\right)$, we have

$$
\begin{aligned}
\mathbb{E}\left[\|\hat{V}_k(t)\hat{V}_k(t)^\intercal - V_k(t)V_k(t)^\intercal\|_2\right] &\leq \frac{2\alpha_t}{\alpha_t - \mathbb{E}\left[\|M_t - \Sigma_t\|_2\right]} - 2 \\
&\leq \frac{\frac{2c_0}{\sqrt{m}}}{\alpha_t - \frac{c_0}{\sqrt{m}}}
\end{aligned}
\tag{41}
$$

Using equation 38 such that $\frac{c_0}{\sqrt{m}} \leq \frac{\alpha_t}{2}$, we have

$$
\mathbb{E}\left[\|\hat{V}_k(t)\hat{V}_k(t)^\intercal - V_k(t)V_k(t)^\intercal\|_2\right] \leq O\left(\frac{G\rho\sqrt{\ln p}\gamma_2(\mathcal{W}, d)}{\alpha_t\sqrt{m}}\right).
\tag{42}
$$

That completes the proof. ∎

**Lemma 2 (Lemma 2.2.3 in Talagrand (2014))** *Consider a r.v. $Y \geq 0$ which satisfies*

$$
\forall u > 0, \mathbb{P}(Y \geq u) \leq A\exp\left(-\frac{u^2}{B^2}\right)
\tag{43}
$$

*for certain numbers $A \geq 2$ and $B > 0$. Then*

$$
\mathbb{E}Y \leq CB\sqrt{\log A},
\tag{44}
$$

*where $C$ denotes a universal constant.*

## A.2 GEOMETRY OF GRADIENTS AND $\gamma_2$ FUNCTIONS.

In this section, we provide more intuitions and explanations of $\gamma_2$ functions. We justify our assumptions about the gradient space and provide more examples of the gradient space structure and the corresponding $\gamma_2$ functions.

At a high level, for a metric space $(M, d)$, $\gamma_2(M, d)$ is related to $\sqrt{\log N(M, d, \epsilon)}$ where $N(M, d, \epsilon)$ is the covering number of $M$ with $\epsilon$ balls with metric $d$, but it is considerably sharper. Such sharpening has happened in two stages in the literature: first, based on *chaining*, which considers an integral over all $\epsilon$ yielding the *Dudley bound*, and subsequently, based on *generic chaining*, which considers a hierarchical covering, developed by Talagrand and colleagues, and which yields the sharpest bounds of this type. The official perspective of generic chaining is to view $\gamma_2(M, d)$ as an upper (and lower) bound on suprema of Gaussian processes indexed on $M$ and with metric $d$ [Theorem 2.4.1 in Talagrand (2014)].

Considering $d$ to be the $\ell_2$ norm distance, $\gamma_2(M, d)$ will be the same order as the Gaussian width of $M$ (Vershynin, 2018), which is a scaled version of the mean width of $M$. Structured sets (of gradients) have small Gaussian widths, e.g., a $L_1$ unit ball in $\mathbb{R}^p$ has a Gaussian width of $O(\sqrt{\log p})$, wheras a $L_2$ unit ball in $\mathbb{R}^p$ has a Gaussian width of $O(\sqrt{p})$.

To utilize the structure of gradients as example shown in Figure 2, instead of focusing on the $\gamma_2(\mathcal{W}, d)$, we can also derive the uniform convergence bound using the measurement $d$ on the set of population gradient $\mathbf{m} = \mathbb{E}_{z \in \mathcal{P}}[\nabla(\mathbf{w}, z)]$. We consider a mapping $f : \mathcal{W} \mapsto \mathcal{M}$ from the parameter space $\mathcal{W}$ to the gradient space $\mathcal{M}$, where $f$ can be considered as $f(\mathbf{w}) = \mathbb{E}_{z \in \mathcal{P}}[\nabla(\mathbf{w}, z)]$. With $\mathbf{m}_t = \mathbb{E}_{z \in \mathcal{P}}[\nabla(\mathbf{w}, z)]$ and $\mathcal{M}$ to be the space of the population gradient $\mathbf{m}_t \in \mathcal{M}$, the pseudo metric $d(\mathbf{w}, \mathbf{w}')$ can be written as $d(\mathbf{w}, \mathbf{w}') = d(f(\mathbf{w}), f(\mathbf{w}')) = d(\mathbf{m}, \mathbf{m}')$ with $\mathbf{m} = \mathbb{E}_{z \in \mathcal{P}}[\nabla(\mathbf{w}, z)]$ and $\mathbf{m}' = \mathbb{E}_{z \in \mathcal{P}}[\nabla(\mathbf{w}', z)]$. With such a mapping $f$, the admissible sequence $\Gamma_{\mathcal{W}} = \{\mathcal{W}_n : n \geq 0\}$ of $\mathcal{W}$ in the proof of Theorem 2 corresponds to the admissible sequence $\Gamma_{\mathcal{M}} = \{\mathcal{M}_n : n \geq 0\}$ of $\mathcal{M}$. The $\gamma_2(\mathcal{W}, d) = \inf_{\Gamma_{\mathcal{W}}} \sup_{\mathbf{w} \in \mathcal{W}} \sum_{n \geq 0} 2^{n/2} d(\mathbf{w}, \mathcal{W}_n) = \inf_{\Gamma_{\mathcal{M}}} \sup_{\mathbf{m} \in \mathcal{M}} \sum_{n \geq 0} 2^{n/2} d(\mathbf{m}, \mathcal{M}_n) = \gamma_2(\mathcal{M}, d)$. Considering $d(\mathbf{m}, \mathbf{m}') = \|\mathbf{m} - \mathbf{m}'\|_2$, the $\gamma_2(\mathcal{M}, d)$ will be the same order as the Gaussian width of $\mathcal{M}$, i.e., $w(\mathcal{M}) = \mathbb{E}_{\mathbf{v}}[\sup_{\mathbf{m} \in \mathcal{M}} \langle \mathbf{m}, \mathbf{v} \rangle]$, where $\mathbf{v} \sim N(0, \mathbb{I}_p)$. Below, we provide more examples of the gradient space structure and the $\gamma_2$ functions.

**Ellipsoid.** The Gaussian width $w(\mathcal{M})$ depends on the structure of the gradient $\mathbf{m}$. In Figure 2, we observe that, for each coordinates of the gradient is of small value along the training trajectory and thus $\mathcal{M}$ includes all gradients living in an ellipsoid, i.e., $\mathcal{M} = \{\mathbf{m}_t \in \mathbb{R}^p \mid \sum_{j=1}^p \mathbf{m}_t(j)^2 / \mathbf{e}(j)^2 \leq 1, \mathbf{e} \in \mathbb{R}^p\}$. Then we have $\gamma_2(\mathcal{M}, d) \leq c_1 w(\mathcal{M}) \leq c_2 \|\mathbf{e}\|_2$ Talagrand (2014), where $c_1$ and $c_2$ are absolute constants. If the elements of $\mathbf{e}$ sorted in decreasing order satisfy $\mathbf{e}(j) \leq c_3 / \sqrt{j}$ for all $j \in [p]$, then $\gamma_2(\mathcal{M}, d) \leq O(\sqrt{\log p})$.

**Composition.** Based on the composition properties of $\gamma_2$ functions Talagrand (2014), one can construct additional examples of the gradient spaces. If $\mathcal{M} = \mathcal{M}_1 + \mathcal{M}_2 = \{\mathbf{m}_1 + \mathbf{m}_2, \mathbf{m}_1 \in \mathcal{M}_1, \mathbf{m}_2 \in \mathcal{M}_2\}$, the Minkowski sum, then $\gamma_2(\mathcal{M}, d) \leq c(\gamma_2(\mathcal{M}_1, d) + \gamma_2(\mathcal{M}_2, d))$ (Theorem 2.4.15 in Talagrand (2014)), where $c$ is an absolute constant. If $\mathcal{M}$ is a union of several subset, i.e., $\mathcal{M} = \cup_{h=1}^D M_h$, then by using an union bound on Theorem 2, we have $\gamma_2(\mathcal{M}, d) \leq \sqrt{\log D} \max_h \gamma_2(\mathcal{M}_h, d)$. Thus, if $\mathcal{M}$ is an union of $D = p^s$ ellipsoids, i.e., polynomial in $p$, then $\gamma_2(\mathcal{M}, \|\cdot\|_2) \leq O(\sqrt{s} \log p)$.

# B PROOFS FOR SECTION 3.2

In this section, we present the proofs for Section 3.2. Then we present the error rate for convex problems in the subsequent section.

**Theorem 4 (Smooth and Non-convex)** *For $\rho$-smooth function $\hat{L}_n(\mathbf{w})$, under Assumptions 1 and 2, let $\Lambda = \frac{\sum_{t=1}^T 1/\alpha_t^2}{T}$, for any $\epsilon, \delta > 0$, with $T = O(n^2 \epsilon^2)$ and $\eta_t = \frac{1}{\sqrt{T}}$, PDP-SGD achieves:*

$$\frac{1}{T} \sum_{t=1}^T \mathbb{E}\|V_k(t) V_k(t)^\mathsf{T} \nabla \hat{L}_n(\mathbf{w}_t)\|_2^2 \leq \tilde{O}\left(\frac{k\rho G^2}{n\epsilon}\right) + O\left(\frac{\Lambda G^4 \rho^2 \gamma_2^2(\mathcal{W}, d) \ln p}{m}\right). \quad (3)$$

*Additionally, assuming the principal component of the gradient dominates, i.e., there exist $c > 0$, such that $\frac{1}{T} \sum_{t=1}^T \|[\nabla \hat{L}_n(\mathbf{w}_t)]^\perp\|_2^2 \leq c \frac{1}{T} \sum_{t=1}^T \|[\nabla \hat{L}_n(\mathbf{w}_t)]^\|\|_2^2$, we have*

$$\mathbb{E}\|\nabla \hat{L}_n(\mathbf{w}_R)\|_2^2 \leq \tilde{O}\left(\frac{k\rho G^2}{n\epsilon}\right) + O\left(\frac{\Lambda G^4 \rho^2 \gamma_2^2(\mathcal{W}, d) \ln p}{m}\right), \quad (4)$$

*where $\mathbf{w}_R$ is uniformly sampled from $\{\mathbf{w}_1, ..., \mathbf{w}_T\}$.*

*Proof:* Recall that $\mathbf{g}_t = \frac{1}{|B_t|} \sum_{z_i \in B_t} \nabla \ell(\mathbf{w}_t, z_i)$. With $B_t$ uniformly sampled from $S$, we have

$$\mathbb{E}_t[\mathbf{g}_t] = \nabla \hat{L}_n(\mathbf{w}_t). \quad (45)$$

Recall that the update of Algorithm 1 is

$$\tilde{\mathbf{g}}_t = \hat{V}_k \hat{V}_k^\mathsf{T} (\mathbf{g}_t + \mathbf{b}_t), \text{ and } \mathbf{w}_{t+1} = \mathbf{w}_t - \eta_t \tilde{\mathbf{g}}_t. \quad (46)$$

Let $\bar{\mathbf{g}}_t = \mathbf{g}_t + \hat{V}_k\hat{V}_k^\mathsf{T}\mathbf{b}_t$, and $\Delta_t = \hat{V}_k\hat{V}_k^\mathsf{T}\mathbf{g}_t - \mathbf{g}_t$. Then we have

$$\tilde{\mathbf{g}}_t = \bar{\mathbf{g}}_t + \Delta_t. \tag{47}$$

Since $\mathbf{b}_t$ is a zero mean Gaussian vector, we have

$$\mathbb{E}_t[\bar{\mathbf{g}}_t] = \mathbf{g}_t. \tag{48}$$

For $\rho$-smooth [5] function $\hat{L}_n(\mathbf{w})$, conditioned on $\mathbf{w}_t$, we have

$$
\begin{aligned}
\mathbb{E}_t\left[\hat{L}_n(\mathbf{w}_{t+1})\right] &\leq \hat{L}_n(\mathbf{w}_t) + \mathbb{E}_t\left[\left\langle \nabla\hat{L}_n(\mathbf{w}_t), \mathbf{w}_{t+1} - \mathbf{w}_t \right\rangle\right] + \frac{\rho}{2}\eta_t^2\mathbb{E}_t\left[\|\tilde{\mathbf{g}}_t\|_2^2\right] \\
&= \hat{L}_n(\mathbf{w}_t) - \eta_t\mathbb{E}_t\left[\left\langle \nabla\hat{L}_n(\mathbf{w}_t), \bar{\mathbf{g}}_t + \Delta_t \right\rangle\right] + \frac{\rho}{2}\eta_t^2\mathbb{E}_t\left[\|\tilde{\mathbf{g}}_t\|_2^2\right] \\
&= \hat{L}_n(\mathbf{w}_t) - \eta_t\left\langle \nabla\hat{L}_n(\mathbf{w}_t), \nabla\hat{L}_n(\mathbf{w}_t) + \mathbb{E}_t[\Delta_t] \right\rangle + \frac{\rho}{2}\eta_t^2\mathbb{E}_t\left[\|\tilde{\mathbf{g}}_t\|_2^2\right] \\
&= \hat{L}_n(\mathbf{w}_t) - \eta_t\left\|\nabla\hat{L}_n(\mathbf{w}_t)\right\|_2^2 - \eta_t\left\langle \nabla\hat{L}_n(\mathbf{w}_t), \mathbb{E}_t[\Delta_t] \right\rangle + \frac{\rho}{2}\eta_t^2\mathbb{E}_t\left[\|\tilde{\mathbf{g}}_t\|_2^2\right]
\end{aligned}
\tag{49}
$$

Rearrange the above inequality, we have

$$\eta_t\left\|\nabla\hat{L}_n(\mathbf{w}_t)\right\|_2^2 + \eta_t\underbrace{\mathbb{E}_t\left\langle \nabla\hat{L}_n(\mathbf{w}_t), \Delta_t \right\rangle}_{D_{t,1}} \leq \hat{L}_n(\mathbf{w}_t) - \mathbb{E}_t\left[\hat{L}_n(\mathbf{w}_{t+1})\right] + \frac{\rho}{2}\eta_t^2\underbrace{\mathbb{E}_t\left[\|\tilde{\mathbf{g}}_t\|_2^2\right]}_{D_{t,2}} \tag{50}$$

For $D_{t,1}$, let

$$\nabla\hat{L}_n(\mathbf{w}_t) = \Pi_{Q_t}[\nabla\hat{L}_n(\mathbf{w}_t)] + \Pi_{Q_t^\perp}[\nabla\hat{L}_n(\mathbf{w}_t)] \tag{51}$$

where $Q_t$ is $\hat{V}_k(t)\hat{V}_k(t)^\mathsf{T}$ and $\Pi_{Q_t}$ is the projection. $Q_t^\perp$ is the null space of $Q_t$.

$$\Pi_{Q_t}[\nabla\hat{L}_n(\mathbf{w}_t)] = \hat{V}_k(t)\hat{V}_k(t)^\mathsf{T}\nabla\hat{L}_n(\mathbf{w}_t) \tag{52}$$

and

$$\Pi_{Q_t^\perp}[\nabla\hat{L}_n(\mathbf{w}_t)] = \left[\mathbb{I} - \hat{V}_k(t)\hat{V}_k(t)^\mathsf{T}\right]\nabla\hat{L}_n(\mathbf{w}_t). \tag{53}$$

We have

$$
\begin{aligned}
\Delta_t &= \Pi_{Q_t}[\hat{V}_k(t)\hat{V}_k(t)^\mathsf{T}\mathbf{g}_t - \mathbf{g}_t] + \Pi_{Q_t^\perp}[\hat{V}_k(t)\hat{V}_k(t)^\mathsf{T}\mathbf{g}_t - \mathbf{g}_t] \\
&= -\Pi_{Q_t^\perp}[\mathbf{g}_t].
\end{aligned}
\tag{54}
$$

So that we have

$$
\begin{aligned}
D_{t,1} &= \mathbb{E}_t\left\langle \nabla\hat{L}_n(\mathbf{w}_t), \Delta_t \right\rangle \\
&= -\mathbb{E}_t\left\langle \Pi_{Q_t}[\nabla\hat{L}_n(\mathbf{w}_t)] + \Pi_{Q_t^\perp}[\nabla\hat{L}_n(\mathbf{w}_t)], \ \Pi_{Q_t^\perp}[\mathbf{g}_t] \right\rangle \\
&= -\mathbb{E}_t\left\langle \Pi_{Q_t^\perp}[\nabla\hat{L}_n(\mathbf{w}_t)], \ \left[\mathbb{I} - \hat{V}_k(t)\hat{V}_k(t)^\mathsf{T}\right]\mathbf{g}_t \right\rangle \\
&= -\left\langle \left[\mathbb{I} - \hat{V}_k(t)\hat{V}_k(t)^\mathsf{T}\right]\nabla\hat{L}_n(\mathbf{w}_t), \left[\mathbb{I} - \hat{V}_k(t)\hat{V}_k(t)^\mathsf{T}\right]\nabla\hat{L}_n(\mathbf{w}_t) \right\rangle \\
&= -\nabla\hat{L}_n(\mathbf{w}_t)^\mathsf{T}\left[\mathbb{I} - \hat{V}_k(t)\hat{V}_k(t)^\mathsf{T}\right]\nabla\hat{L}_n(\mathbf{w}_t).
\end{aligned}
\tag{55}
$$

Bringing the above to equation 50, the right-hand side of equation 50 becomes

$$
\begin{aligned}
\eta_t\|\nabla\hat{L}_n(\mathbf{w}_t)\|_2^2 + \eta_t D_{t,1} &= \eta_t\nabla\hat{L}_n(\mathbf{w}_t)^\mathsf{T}[\mathbb{I}]\nabla\hat{L}_n(\mathbf{w}_t) - \eta_t\nabla\hat{L}_n(\mathbf{w}_t)^\mathsf{T}\left[\mathbb{I} - \hat{V}_k(t)\hat{V}_k(t)^\mathsf{T}\right]\nabla\hat{L}_n(\mathbf{w}_t) \\
&= \eta_t\nabla\hat{L}_n(\mathbf{w}_t)^\mathsf{T}\left[\hat{V}_k(t)\hat{V}_k(t)^\mathsf{T}\right]\nabla\hat{L}_n(\mathbf{w}_t) \\
&= \eta_t\|\hat{V}_k(t)\hat{V}_k(t)^\mathsf{T}\nabla\hat{L}_n(\mathbf{w}_t)\|_2^2.
\end{aligned}
\tag{56}
$$

---

[5] The Assumption 2 suggests that the $\hat{L}_n(\mathbf{w})$ is $\rho$-smooth.

So that we have

$$\eta_t \|\hat{V}_k(t)\hat{V}_k(t)^\mathsf{T}\nabla\hat{L}_n(\mathbf{w}_t)\|_2^2 \le \hat{L}_n(\mathbf{w}_t) - \mathbb{E}_t\left[\hat{L}_n(\mathbf{w}_{t+1})\right] + \frac{\rho}{2}\eta_t^2 \underbrace{\mathbb{E}_t\left[\|\tilde{\mathbf{g}}_t\|_2^2\right]}_{D_{t,2}}. \tag{57}$$

For $D_{t,2}$, we have

$$\mathbb{E}_t\|\tilde{\mathbf{g}}_t\|_2^2 = \mathbb{E}_t\left[\left\|\hat{V}_k\hat{V}_k^T\mathbf{g}_t + \hat{V}_k\hat{V}_k^T\mathbf{b}_t\right\|_2^2\right]$$

$$= \mathbb{E}_t\left[\left\|\hat{V}_k\hat{V}_k^T\mathbf{g}_t\right\|_2^2 + \left\|\hat{V}_k\hat{V}_k^T\mathbf{b}_t\right\|_2^2\right]$$

$$\le G^2 + k\sigma^2. \tag{58}$$

Thus,

$$D_{t,2} = \mathbb{E}_t\left[\|\bar{\mathbf{g}}_t\|_2^2 + \|\Delta_t\|_2^2\right] \le G^2 + k\sigma^2. \tag{59}$$

Bringing the upper bound of $D_{t,2}$ to equation 57, setting $\eta_t = \frac{1}{\sqrt{T}}$, using telescoping sum and taking the expectation over all iterations, we have

$$\frac{1}{T}\sum_{t=1}^T \mathbb{E}\|\hat{V}_k(t)\hat{V}_k(t)^\mathsf{T}\nabla\hat{L}_n(\mathbf{w}_t)\|_2^2 \le \frac{\hat{L}_n(\mathbf{w}_1) - \hat{L}_n^\star}{\sqrt{T}} + \frac{\rho(G^2 + k\sigma^2)}{2\sqrt{T}} \tag{60}$$

With triangle inequality and Theorem 3, we have

$$\left\|V_k(t)V_k(t)^\mathsf{T}\nabla\hat{L}_n(\mathbf{w}_t)\right\|_2^2 \le 2\left\|\hat{V}_k(t)\hat{V}_k(t)^\mathsf{T}\nabla\hat{L}_n(\mathbf{w}_t)\right\|_2^2 + 2\left\|\left(V_k(t)V_k(t)^\mathsf{T} - \hat{V}_k(t)\hat{V}_k(t)^\mathsf{T}\right)\nabla\hat{L}_n(\mathbf{w}_t)\right\|_2^2$$

$$\le 2\left\|\hat{V}_k(t)\hat{V}_k(t)^\mathsf{T}\nabla\hat{L}_n(\mathbf{w}_t)\right\|_2^2 + O\left(\frac{G^4\rho^2\ln p\gamma_2^2(\mathcal{W}, d)}{\alpha_t^2 m}\right). \tag{61}$$

With equation 60, we have

$$\frac{1}{T}\sum_{t=1}^T \mathbb{E}\|V_k(t)V_k(t)^\mathsf{T}\nabla\hat{L}_n(\mathbf{w}_t)\|_2^2 \le \frac{\hat{L}_n(\mathbf{w}_1) - \hat{L}_n^\star}{\sqrt{T}/2} + \frac{\rho(G^2 + k\sigma^2)}{\sqrt{T}} + O\left(\frac{\Lambda G^4 L\rho^2\gamma_2^2(\mathcal{W}, d)\ln p}{m}\right). \tag{62}$$

where $\Lambda = \sum_{t=1}^T \frac{1}{\alpha_t^2}$.

Let $\left[\nabla\hat{L}_n(\mathbf{w}_t)\right]^\| = V_k(t)V_k(t)^\mathsf{T}\nabla\hat{L}_n(\mathbf{w}_t)$ and $\left[\nabla\hat{L}_n(\mathbf{w}_t)\right]^\perp = \nabla\hat{L}_n(\mathbf{w}_t) - V_k(t)V_k(t)^\mathsf{T}\nabla\hat{L}_n(\mathbf{w}_t)$.

Assuming there exist $c > 0$, we have

$$\sum_{t=1}^T \mathbb{E}\left\|\left[\nabla\hat{L}_n(\mathbf{w}_t)\right]^\perp\right\|_2^2 \le c\sum_{t=1}^T \mathbb{E}\left\|\left[\nabla\hat{L}_n(\mathbf{w}_t)\right]^\|\right\|_2^2. \tag{63}$$

Then we have

$$\frac{1}{T}\sum_{t=1}^T \mathbb{E}\|\nabla\hat{L}_n(\mathbf{w}_t)\|_2^2 \le (1+c)\left(\frac{\hat{L}_n(\mathbf{w}_1) - \hat{L}_n^\star}{\sqrt{T}/2} + \frac{\rho(G^2 + k\sigma^2)}{\sqrt{T}} + O\left(\frac{\Lambda G^4 L\rho^2\gamma_2^2(\mathcal{W}, d)\ln p}{m}\right)\right). \tag{64}$$

Take $T = n^2\epsilon^2$, with $\mathbb{E}\left[\|\nabla\hat{L}_n(\mathbf{w}_R)\|_2^2\right] = \frac{1}{T}\sum_{t=1}^T \mathbb{E}\|\nabla\hat{L}_n(\mathbf{w}_t)\|_2^2$, we have

$$\mathbb{E}\|\nabla\hat{L}_n(\mathbf{w}_R)\|_2^2 \le \tilde{O}\left(\frac{k\rho G^2}{n\epsilon}\right) + O\left(\frac{\Lambda G^4\rho^2\gamma_2^2(\mathcal{W}, d)\ln p}{m}\right), \tag{65}$$

where $\mathbf{w}_R$ is uniformly sampled from $\{\mathbf{w}_1, ..., \mathbf{w}_T\}$. ∎

## C    ERROR RATE OF CONVEX PROBLEMS

For the convex and Lipschitz functions, we consider the low-rank structure of the gradient space, i.e, the population gradient second momment $\Sigma_t$ is of rank-$k$, which is a special case of the principal gradient dominate assumption when $\|[\nabla \hat{L}_n(\mathbf{w}_t)]^\perp\|_2 = 0$.

**Theorem 6 (Convex and Lipschitz)** *For G-Lipschitz and convex function $\hat{L}_n(\mathbf{w})$, under Assumptions 1,2 and assuming $\Sigma_t$ is of rank-k, let $\Lambda = \frac{\sum_{t=1}^T 1/\alpha_t}{T}$, for any $\epsilon, \delta > 0$, with $T = O(n^2\epsilon^2)$, step size $\eta_t = \frac{1}{\sqrt{T}}$, the PDP-SGD achieves*

$$\mathbb{E}\left[\hat{L}_n(\bar{\mathbf{w}})\right] - \hat{L}_n(\mathbf{w}^\star) \le O\left(\frac{kG^2}{n\epsilon}\right) + O\left(\frac{\Lambda G \rho \gamma_2(\mathcal{W}, d) \ln p}{\sqrt{m}}\right), \tag{66}$$

*where $\bar{\mathbf{w}} = \frac{\sum_{t=1}^T \mathbf{w}_t}{T}$, and $\mathbf{w}^\star$ is the minima of $\hat{L}_n(\mathbf{w})$.*

PDP-SGD also demonstrates an improvement from a factor of $p$ to $k$ compared to the error rate of DP-SGD for convex functions (Bassily et al., 2014; 2019a). PDP-SGD also has the subspace reconstruction error, depending on the $\gamma_2$ function and eigen-gap term $\Lambda = \sum_{t=1}^T 1/\alpha_t/T$. From previous discussions, the $\gamma_2$ is a constant with suitable assumptions of the gradient structure. For the eigen-gap term, if $\alpha_t$ stays as a constant in the training procedure as shown by Figure 1, $\Lambda$ will be a constant and the bound scales logarithmically with $p$. If one assumes the eigen-gap $\alpha_t$ decays as training proceed, e.g., $\alpha_t = \frac{1}{t^{1/2}}$ for $t > 0$, then we have $\Lambda = O(\sqrt{T})$. In this case, with $T = O(n^2\epsilon^2)$, PDP-SGD requires public data size $m = O(n\epsilon)$.

Recently, Kairouz et al. (2020) propose a noisy version of AdaGrad algorithm for unconstrained convex empirical risk minimization. Their algorithm operates with only private data and also achieves a dimension-free excess risk bound, i.e., $\tilde{O}(r/n\epsilon)$, by assuming the gradients along the path of optimization lie in a low-dimensional subspace with constant rank $r$. The bounds are not directly comparable since the assumptions in Kairouz et al. (2020) are different from those in this paper, i.e., Kairouz et al. (2020) assume that the accumulated gradients along the training process lie in a constant subspace which requires the rank of the gradient space does not explode when adding more stochastic gradients. Our work does not impose such a constant subspace assumption on $\Sigma_t$ allowing the subspace of $\Sigma_t$ to be different along the training process. In other words, our bounds hold even when the rank of the accumulated stochastic gradients space increases as training proceeds.

*Proof of Theorem 6*: By the convexity of $\hat{L}_n(\mathbf{w})$, we have

$$\hat{L}_n(\mathbf{w}_t) - \hat{L}_n(\mathbf{w}^\star) \le \langle \mathbf{w}_t - \mathbf{w}^\star, \nabla \hat{L}_n(\mathbf{w}_t) \rangle. \tag{67}$$

At iteration $t$, we have $\mathbf{w}_{t+1} = \mathbf{w}_t - \eta_t \tilde{\mathbf{g}}_t$, where $\tilde{\mathbf{g}}_t = \hat{V}_k \hat{V}_k^\mathsf{T} \mathbf{g}_t + \hat{V}_k \hat{V}_k^\mathsf{T} \mathbf{b}_t$.

Let $\bar{\mathbf{g}}_t = \mathbf{g}_t + \hat{V}_k \hat{V}_k^\mathsf{T} \mathbf{b}_t$, and $\Delta_t = \hat{V}_k \hat{V}_k^\mathsf{T} \mathbf{g}_t - \mathbf{g}_t$. Then we have

$$\tilde{\mathbf{g}}_t = \bar{\mathbf{g}}_t + \Delta_t. \tag{68}$$

Recall that $\mathbf{g}_t = \frac{1}{|B_t|} \sum_{z_i \in B_t} \nabla \ell(\mathbf{w}_t, z_i)$. With $B_t$ uniformly sampled from $S$, we have

$$\mathbb{E}_t[\mathbf{g}_t] = \nabla \hat{L}_n(\mathbf{w}_t). \tag{69}$$

Since $\mathbf{b}_t$ is a zero mean Gaussian vector, we have $\mathbb{E}_t[\bar{\mathbf{g}}_t] = \mathbf{g}_t$.

By convexity, conditioned at $\mathbf{w}_t$, we have

$$
\begin{aligned}
&\mathbb{E}_t[\hat{L}_n(\mathbf{w}_t) - \hat{L}_n(\mathbf{w}^\star)] \\
&\leq \mathbb{E}_t\langle \mathbf{w}_t - \mathbf{w}^\star, \mathbf{g}_t\rangle \\
&= \mathbb{E}_t\langle \mathbf{w}_t - \mathbf{w}^\star, \bar{\mathbf{g}}_t\rangle \\
&= \frac{1}{\eta_t}\mathbb{E}_t\langle \mathbf{w}_t - \mathbf{w}^\star, \mathbf{w}_t - \mathbf{w}_{t+1} - \eta_t\Delta_t\rangle \\
&= \frac{1}{2\eta_t}\mathbb{E}_t\left(\|\mathbf{w}_t - \mathbf{w}^\star\|_2^2 + \eta_t^2\|\bar{\mathbf{g}}_t\|_2^2 - \|\mathbf{w}_{t+1} - \mathbf{w}^\star - \eta_t\Delta_t\|_2^2\right) \\
&= \frac{1}{2\eta_t}\mathbb{E}_t\left(\|\mathbf{w}_t - \mathbf{w}^\star\|_2^2 + \eta_t^2\|\bar{\mathbf{g}}_t\|_2^2 - \|\mathbf{w}_{t+1} - \mathbf{w}^\star\|_2^2 - \eta_t^2\|\Delta_t\|_2^2 + 2\langle\mathbf{w}_{t+1} - \mathbf{w}^\star, \eta_t\Delta_t\rangle\right) \\
&= \frac{1}{2\eta_t}\mathbb{E}_t\left(\|\mathbf{w}_t - \mathbf{w}^\star\|_2^2 - \|\mathbf{w}_{t+1} - \mathbf{w}^\star\|_2^2\right) + \frac{\eta_t}{2}\mathbb{E}_t[\|\bar{\mathbf{g}}_t\|_2^2] - \frac{\eta_t}{2}\mathbb{E}_t[\|\Delta_t\|_2^2] \\
&\quad + \mathbb{E}_t\langle\mathbf{w}_{t+1} - \mathbf{w}^\star, \Delta_t\rangle \\
&\overset{(a)}{\leq} \frac{1}{2\eta_t}\mathbb{E}_t\left(\|\mathbf{w}_t - \mathbf{w}^\star\|_2^2 - \|\mathbf{w}_{t+1} - \mathbf{w}^\star\|_2^2\right) + \frac{\eta_t}{2}\left(G^2 + k\sigma^2\right) + B\mathbb{E}_t\|\Delta_t\|_2, \quad (70)
\end{aligned}
$$

where $(a)$ is true since

$$
\begin{aligned}
\mathbb{E}_t\|\bar{\mathbf{g}}_t\|_2^2 &\leq \mathbb{E}_t[\|\mathbf{g}_t\|_2^2 + \|\hat{V}_k\hat{V}_k^{\mathsf{T}}\mathbf{b}_t\|_2^2] \\
&\leq G^2 + k\sigma^2, \quad (71)
\end{aligned}
$$

and $\|\mathbf{w}_{t+1} - \mathbf{w}^\star\|_2 \leq B$ [6].

Let $\eta_t = \frac{1}{\sqrt{T}}$, taking the expectation over all iterations and sum over $t = 1, .., T$, we have

$$
\frac{1}{T}\mathbb{E}\left[\sum_{t=1}^{\mathsf{T}}\hat{L}_n(\mathbf{w}_t) - \hat{L}_n(\mathbf{w}^\star)\right] \leq \frac{\|\mathbf{w}_0 - \mathbf{w}^\star\|_2^2 + G^2 + k\sigma^2}{2\sqrt{T}} + \frac{B\sum_{t=1}^{\mathsf{T}}\mathbb{E}_t\|\Delta_t\|_2}{T}. \quad (72)
$$

From Theorem 3, we have

$$
\begin{aligned}
\mathbb{E}_t\left[\|\Delta_t\|_2\right] &= \mathbb{E}\left[\left\|\hat{V}_k(t)\hat{V}_k(t)^{\mathsf{T}}\mathbf{g}_t - V(t)V(t)^{\mathsf{T}}\mathbf{g}_t\right\|_2\right] \\
&\leq \mathbb{E}\left[\left\|\hat{V}_k(t)\hat{V}_k(t)^{\mathsf{T}} - V(t)V(t)^{\mathsf{T}}\right\|_2 G\right] \\
&\leq G\mathbb{E}\left[\left\|\hat{V}_k(t)\hat{V}_k(t)^{\mathsf{T}} - V_k(t)V_k(t)^{\mathsf{T}}\right\|_2 + \|V_k(t)V_k(t)^{\mathsf{T}} - V(t)V(t)^{\mathsf{T}}\|_2\right] \\
&\leq O\left(\frac{G\rho\ln p\gamma_2(\mathcal{W}, d)}{\alpha_t\sqrt{m}}\right), \quad (73)
\end{aligned}
$$

where the last inequality holds because the $\Sigma_t$ is of rank $k$ and $V(t) = V_k(t)$.

Bring this to equation 72, use the fact that $\|\mathbf{w}_0 - \mathbf{w}^\star\| < B$, with Jensen's inequality we have

$$
\mathbb{E}\left[\hat{L}_n(\bar{\mathbf{w}})\right] - \hat{L}_n(\mathbf{w}^\star) \leq \frac{B^2 + G^2 + k\sigma^2}{2\sqrt{T}} + O\left(\frac{\Lambda G\rho\ln p\gamma_2(\mathcal{W}, d)}{\sqrt{m}}\right). \quad (74)
$$

where $\Lambda = \sum_{t=1}^{\mathsf{T}}\frac{1}{\alpha_t}$.

With $\sigma^2 = \frac{G^2 T}{n^2\epsilon^2}$, let $T = n^2\epsilon^2$, we have

$$
\mathbb{E}\left[\hat{L}_n(\bar{\mathbf{w}})\right] - \hat{L}_n(\mathbf{w}^\star) \leq O\left(\frac{kG^2 B^2}{n\epsilon}\right) + O\left(\frac{\Lambda G\rho\ln p\gamma_2(\mathcal{W}, d)}{\sqrt{m}}\right). \quad (75)
$$

That completes the proof. ∎

---

[6] For convex problem, we consider $\mathbf{w} \in \mathcal{H}$, where $\mathcal{H} = \{\mathbf{w} : \|\mathbf{w}\| \leq B\}$.

## D    EXPERIMENTAL SETUP AND ADDITIONAL RESULTS

**Datasets and Network Structure.** The MNIST and Fashion MNIST datasets both consist of 60,000 training examples and 10,000 test examples. To construct the private training set, we randomly sample 10,000 samples from the original training set of MNIST, then we randomly sample 100 samples from the rest to construct as the public dataset [7]. Details refer to Table 2. For both MNIST and Fashion MNIST, we use a convolutional neural network that follows the structure in Papernot et al. (2020) whose architecture is described in Table 1. All experiments have been run on NVIDIA Tesla K40 GPUs.

Table 1: Network architecture for MNIST and Fashion MNIST.

| Layer | Parameters |
|---|---|
| Convolution | 16 filters of $8 \times 8$, strides 2 |
| Max-Pooling | $2 \times 2$ |
| Convolution | 32 filters of $4x4$, strides 2 |
| Max-Pooling | $2 \times 2$ |
| Fully connected | 32 units |
| Softmax | 10 units |

Table 2: Neural network and datasets setup.

| Dataset | Model | features | classes | Training size | Public size | Test size |
|---|---|---|---|---|---|---|
| MNIST | CNN | $28 \times 28$ | 10 | 10,000 | 100 | 10,000 |
| Fashion MNIST | CNN | $28 \times 28$ | 10 | 10,000 | 100 | 10,000 |

**Hyper-parameter Setting.** We consider different choices of the noise scale, i.e., $\sigma = \{18, 14, 10, 8, 6, 4\}$ for MNIST and $\sigma = \{18, 14, 10, 6, 4, 2\}$ for Fashion MNIST. Cross-entropy is used as our loss function throughout experiments. The mini-batch size is set to be 250 for both MNIST and Fashion MNIST. For the step size, we follow the grid search method with search space $\{0.01, 0.05, 0.1, 0.2\}$ to tune the step size for MNIST and the search space is $\{0.01, 0.02, 0.05, 0.1, 0.2\}$ for Fashion MNIST. We choose the step size based on the training accuracy at the last epoch. The best step sizes for DP-SGD and PDP-SGD for different privacy levels are presented in Table 3 and Table 4 for MNIST and Fashion MNIST, respectively. For training, a fixed budget on the number of epochs i.e., 30 is assigned for the each task. We repeat each experiments 3 times and report the mean and standard deviation of the accuracy on the training and test set. For PDP-SGD, the projection dimension $k$ is a hyper-parameter and it illustrates a trade-off between the reconstruction error and the noise reduction. A small $k$ implies more noise amount will be reduced, and a larger reconstruction error will be introduced. We explored $k = \{20, 30, 50\}$ for MNIST and $k = \{30, 50, 70\}$ for Fashion MNIST, and we found that $k = 50$ and $k = 70$ achieve the best performance for MNIST and Fashion MNIST respectively among the search space we consider. Instead of doing the projection for all epochs, we also explored a start point for the projection, i.e., executing the projection from the 1-th epoch, 15-th epoch. The information of projection dimension $k$ and the starting epoch for projection are also given in Table 3 and Table 4 for MNIST and Fashion MNIST, respectively.

**Privacy Parameter Setting.** Since gradient norm bound $G$ is unknow for deep learning, we follow the gradient clipping method in Abadi et al. (2016) to guarantee the privacy. We implement the micro-batch clipping method in PyTorch [8]. We use micro-batch = 1 and micro-batch = 5 for MNIST and Fashion MNIST, respectively. Note that training with micro-batch clipping will need the noise scaled by micro-batch size to guarantee the same privacy. But it takes less time than training with per-sample clipping, i.e., micro-batch = 1. We follow the Moment Accountant (MA) method (Bu et al., 2019) to calculate the accumulated privacy cost, which depends on the number of epochs, the batch size, $\delta$, and noise variance $\sigma$. With 30 epochs, batch size 250,

---

[7]PDP-SGD can work with larger training set as well. We randomly sample 10,000 samples due to the limitation of computation resources, e.g., GPUs.

[8]We implement the clipping method based on this repository: https://github.com/ChrisWaites/pyvacy

Table 3: Hyper-parameter settings for DP-SGD and PDP-SGD for MNIST.

| | $\sigma = 18$ ($\epsilon = 0.23$) | $\sigma = 14$ ($\epsilon = 0.30$) | $\sigma = 10$ ($\epsilon = 0.42$) | $\sigma = 8$ ($\epsilon = 0.53$) | $\sigma = 6$ ($\epsilon = 0.72$) | $\sigma = 4$ ($\epsilon = 1.09$) |
|---|---|---|---|---|---|---|
| | DP-SGD | | | | | |
| Step size | 0.05 | 0.05 | 0.05 | 0.05 | 0.1 | 0.1 |
| | PDP-SGD | | | | | |
| Step size | 0.1 | 0.2 | 0.2 | 0.1 | 0.1 | 0.2 |
| Starting epoch for projection | 1 | 1 | 1 | 15 | 15 | 15 |
| Projection dimension $k$ | 50 | 50 | 50 | 50 | 50 | 50 |

$10,000$ training samples, and fixing $\delta = 10^{-5}$, the $\epsilon$ is $\{2.41, 1.09, 0.72, 0.42, 0.30, 0.23\}$ for $\sigma \in \{2, 4, 6, 10, 14, 18\}$ for Fashion MNIST. For MNIST, $\epsilon$ is $\{1.09, 0.72, 0.53, 0.42, 0.30, 0.23\}$ corresponding to $\sigma = \{4, 6, 8, 10, 14, 18\}$. Note that the $\epsilon$ presented in this paper is w.r.t. a subset i.e., $10,000$ samples from MNIST and Fashion MNIST. Also, one can fix the value of $\epsilon$ and do a search over the epochs, batch size and noise scale to boost the performance for a fixed privacy level $\epsilon$. We omit such a complicated hyper-parameter tuning since it has a high risk of privacy leakage.

Table 4: Hyper-parameter settings for DP-SGD and PDP-SGD for Fashion MNIST.

| | $\sigma = 18$ ($\epsilon = 0.23$) | $\sigma = 14$ ($\epsilon = 0.30$) | $\sigma = 10$ ($\epsilon = 0.42$) | $\sigma = 6$ ($\epsilon = 0.72$) | $\sigma = 4$ ($\epsilon = 1.09$) | $\sigma = 2$ ($\epsilon = 2.41$) |
|---|---|---|---|---|---|---|
| | DP-SGD | | | | | |
| Step size | 0.01 | 0.01 | 0.01 | 0.02 | 0.02 | 0.05 |
| | PDP-SGD | | | | | |
| Step size | 0.01 | 0.01 | 0.02 | 0.02 | 0.02 | 0.05 |
| Starting epoch for projection | 15 | 15 | 15 | 15 | 15 | 15 |
| Projection dimension $k$ | 70 | 70 | 70 | 70 | 70 | 70 |

**Additional Experimental Results.** Training dynamics of DP-SGD and PDP-SGD with different privacy levels are presented in Figure 7 and Figure 8 respectively for MNIST and Fashion MNIST. The results suggest that for small $\epsilon$, PDP-SGD can effeciently reduce the noise variance injected to the gradient, which improves the training and test accuracy over DP-SGD.

In order to understand the role of projection dimension $k$, we run PDP-SGD with projection starting from the first epoch. Figure 9 reports the PDP-SGD with $k \in \{10, 20, 30, 50\}$ for MNIST with $\epsilon = 0.30$ (Figure 9(a)) and $\epsilon = 0.53$ (Figure 9(b)). Among the choice of $k$, we can see that PDP-SGD with $k = 50$ performs better that the others in terms of the training and test accuracy. PDP-SGD with $k = 10$ proceeds slower than PDP-SGD with $k = 20$ and $k = 50$. This is due to the larger reconstruction error introduced by projecting the gradient to a much smaller subspace, i..e, $k = 10$. However, compared to the gradient dimension $p \approx 25,000$, it is impressive that PDP-SGD with $k = 50$ which projects the gradient to the a much smaller subspace, can achieve better accuracy than DP-SGD.

We also empirically evaluate the effect of the public sample size $m$. Figure 11(a) and Figure 11(b) present the training and test accuracy for PDP-SGD with $m \in \{50, 150, 200\}$ for $\epsilon = 0.23$ and $\epsilon = 1.09$ for Fashion MNIST dataset. The training and test accuracy of PDP-SGD increases as the public sample size increases from 50 to 150. This is consistent with the theoretical analysis that increasing $m$ helps reducing the subspace reconstruction error as suggested by the theoretical bound. Also, PDP-SGD with $m = 150$ and $m = 200$ performs slightly better that $m = 50$ in terms of the training and test accuracy. The results suggest that while a small amount of public datasets are not sufficient for training an accurate predictor, they provide useful gradient subspace projection and accuracy improvement over DP-SGD.

We also compare PDP-SGD and DP-SGD for different number of training samples, i.e., MNIST with 20,000 samples (Figure 12(a)) and Fashion MNIST with 50,000 samples (Figure 12(a)) (100

public samples for both case). The observation that PDP-SGD outperforms DP-SGD for small $\epsilon$ regime in Figure 3 also holds for other number of training samples.

We also explore PDP-SGD with sparse eigen-space computation, i.e., update the projector every $s$ iterates. Note that PDP-SGD with $s = 1$ means computing the top eigen-space at every iteration. Figure 13 reports PDP-SGD with $s = \{1, 10, 20\}$ for (a) MNIST with 50,000 samples and (b) Fashion MNIST with 50,000 samples showing that there is a mild decay for PDP-SGD with fewer eigen-space computation. PDP-SGD with a reduced eigen-space computation also improves the accuracy over DP-SGD.

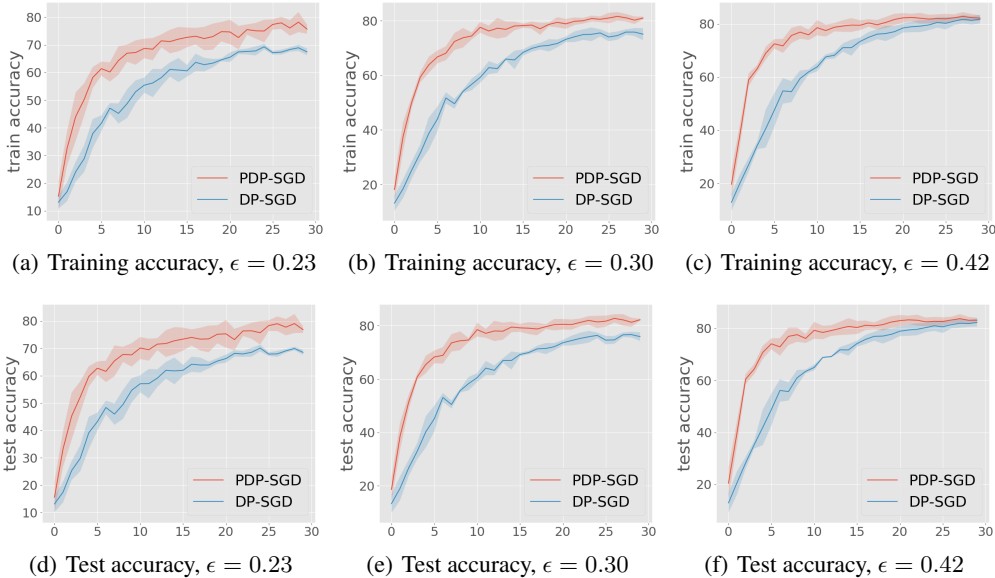

(a) Training accuracy, $\epsilon = 0.23$    (b) Training accuracy, $\epsilon = 0.30$    (c) Training accuracy, $\epsilon = 0.42$

(d) Test accuracy, $\epsilon = 0.23$    (e) Test accuracy, $\epsilon = 0.30$    (f) Test accuracy, $\epsilon = 0.42$

Figure 7: Comparison of DP-SGD and PDP-SGD for MNIST. (a-c) report the training accuracy and (d-f) report the test accuracy for $\epsilon = \{0.23, 0.30, 0.42\}$. The X-axis is the number of epochs, and the Y-axis is the train/test accuracy. DPD-SGD outperforms DP-SGD for small $\epsilon$.

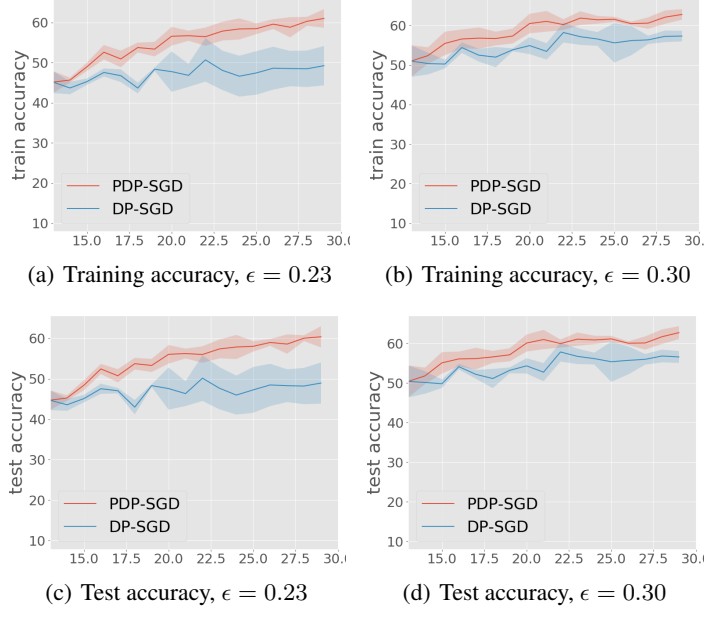

(a) Training accuracy, $\epsilon = 0.23$    (b) Training accuracy, $\epsilon = 0.30$

(c) Test accuracy, $\epsilon = 0.23$    (d) Test accuracy, $\epsilon = 0.30$

Figure 8: Comparison of DP-SGD and PDP-SGD for Fashion MNIST. (a-b) report the training accuracy and (c-d) report the test accuracy for $\epsilon = \{0.23, 0.30\}$. Learning rare is 0.01 for both PDP-SGD and DP-SGD. PDP-SGD starts projection at 15-th epoch. The X-axis is the number of epochs, and the Y-axis is the train/test accuracy. DPD-SGD outperforms DP-SGD for small $\epsilon$.

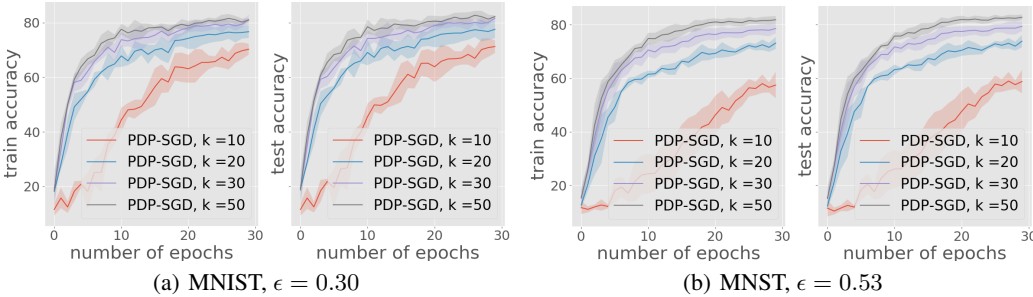

(a) MNIST, $\epsilon = 0.30$            (b) MNST, $\epsilon = 0.53$

Figure 9: Training accuracy and test accuracy for PDP-SGD with $k = \{10, 20, 30, 50\}$ for (a) MNIST with $\epsilon = 0.30$; (b) MNIST with $\epsilon = 0.53$. The X-axis and Y-axis refer to Figure 4. PDP-SGD with $k = 50$ performs better that the others in terms of the training and test accuracy.

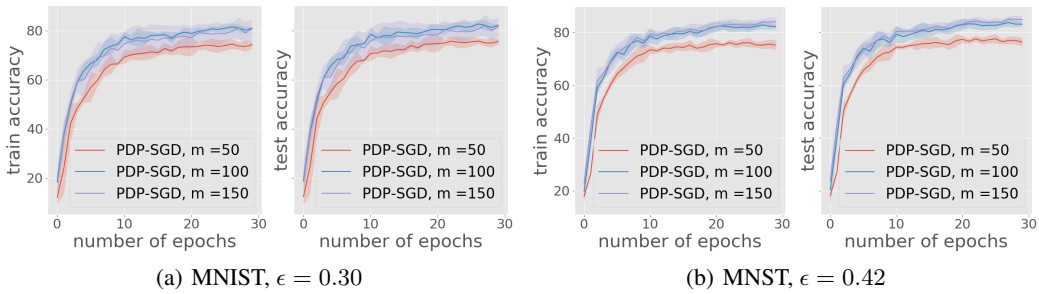

(a) MNIST, $\epsilon = 0.30$            (b) MNST, $\epsilon = 0.42$

Figure 10: Training accuracy and test accuracy for PDP-SGD with $m = \{50, 100, 150\}$ for (a) MNIST with $\epsilon = 0.30$; (b) MNIST with $\epsilon = 0.53$. The X-axis and Y-axis refer to Figure 4. PDP-SGD with $m = 150$ and $m = 100$ perform better that the others in terms of the training and test accuracy.

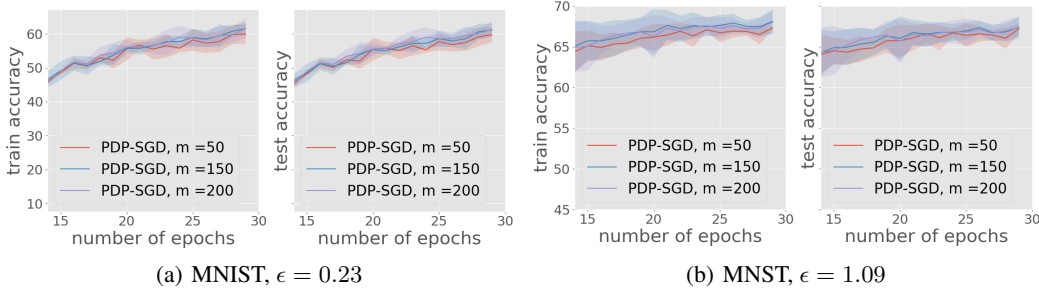

(a) MNIST, $\epsilon = 0.23$            (b) MNST, $\epsilon = 1.09$

Figure 11: Training accuracy and test accuracy for PDP-SGD with $m = \{50, 150, 200\}$ for (a) MNIST with $\epsilon = 0.23$; (b) MNIST with $\epsilon = 0.43$. The X-axis and Y-axis refer to Figure 4. PDP-SGD with $m = 150$ and $m = 200$ performs slightly better that the other one in terms of the training and test accuracy.

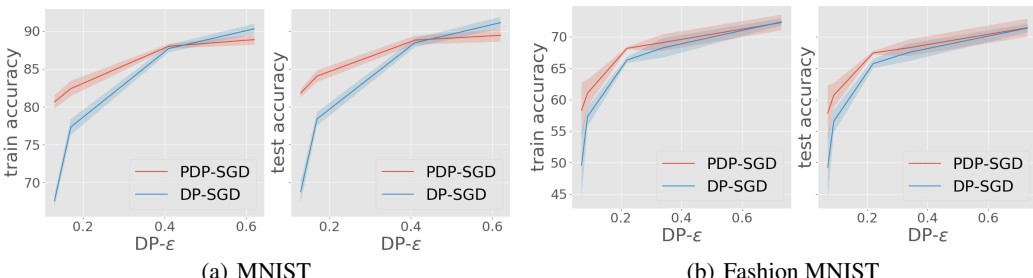

(a) MNIST            (b) Fashion MNIST

Figure 12: Training and test accuracy for DP-SGD and PDP-SGD with different privacy levels for (a) MNIST with 20,000 samples and (b) Fashion MNIST with 50,000 samples. The X-axis and Y-axis refer to Figure 3. For small privacy loss $\epsilon$, PDP-SGD outperforms DP-SGD.

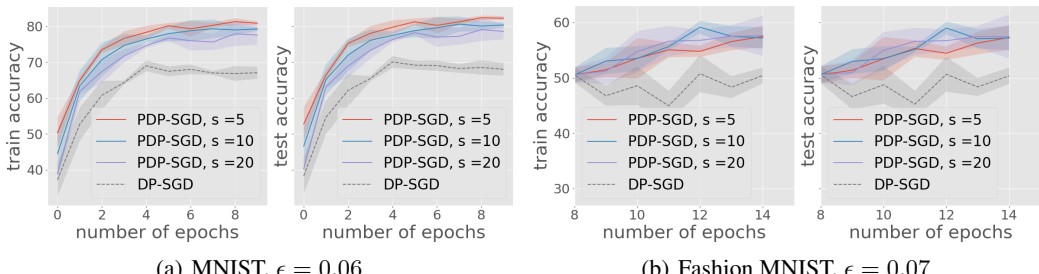

(a) MNIST, $\epsilon = 0.06$              (b) Fashion MNIST, $\epsilon = 0.07$

Figure 13: Training and test accuracy for PDP-SGD with different frequency of eigen-space computation for (a) MNIST with 50,000 samples and (b) Fashion MNIST with 50,000 samples. $s = \{5, 10, 20\}$ is the frequency of subspace update, i.e., compute the eigen-space every $s$ iterates. The X-axis and Y-axis refer to Figure 4. PDP-SGD with a reduced eigen-space computation also improves the accuracy over DP-SGD.

