# OpenReview forum: "Bypassing the Ambient Dimension: Private SGD with Gradient Subspace Identification"
_ICLR.cc/2021/Conference — ICLR 2021 Poster_

### Official Review · AnonReviewer3 · 2020-10-26
**Good theorem but experiments limit**

**Rating:** 7
**Confidence:** 3

**Review:**

The paper considers the problem of solving differentially private empirical risk minimization. To reduce the dependence on dimensionality $p$, they propose Projected DP-SGD (PDP-SGD) that projects the noisy gradients to a low-dimensional subspace computed from a free public dataset at each iteration. They prove that PDP-SGD is differentially private and has only a logarithmic dependence on $p$.

Pros:
1. The paper is well written and easy to follow.
2. It is very appreciated that authors analyze both convex and non-convex cases. All proof seems correct except for a minor mistake mentioned in the next part. What’s more, The method of uniformly bounding $\|M_t-\Sigma_t\|_2$ by generic chaining techniques may be an independent interest.

Cons and some discussion:
1. The establishment of (28) requires $u$ is larger than a constant (like 3), even though this would not affect the correctness of Theorem 2. The last inequality of (55) should be equality.

The main concern of mine is that the experiment part is not so good as the theoretical part.

2. The authors propose to reduce dimensionality by projection. The projected space varies for each iteration and is computed from a public dataset. However, I think the author didn’t explain well the necessity of using an extra dataset. If reducing dimensionality is the most important, why not just use a random projection. [1] shows that a Johnson-Lindenstrauss transform preserves differential privacy, so it is natural to use random projection as a baseline (at least in experiments). However, the baseline considered in experiments excludes this easiest method. Besides, it is better to add non-private SGD as a baseline, which would help illustrate how the proposed method degrades accuracy.
3. The proposed method has a huge space complexity and computation complexity as it requires to formulate $M_t \in \mathbb{R}^{p \times p}$ and compute its top-$k$ eigenvectors at each iteration. It will require $O(p^2)$ space and $O(p^3)$ times. As a remedy, one may hope to reduce the frequency of eigenspace computation. It seems to make sense, since when $w_t$ starts to converge, the difference between consecutive $w_t$ is so small that the difference between consecutive $V(w_t)$ is also small, implying we can reuse the eigenspace estimated in the last iteration. It strikes me that it is better to explore such a heuristic method in experiments to further illustrate the usefulness of the proposed method.
4. I think more datasets should be considered. The author only considered two image datasets. Real datasets like CIFAR can be considered.

[1] Blocki, Jeremiah, et al. "The johnson-lindenstrauss transform itself preserves differential privacy." 2012 IEEE 53rd Annual Symposium on Foundations of Computer Science. IEEE, 2012.



------------------------------------------------------------------------------------------
I have read the authors’ rebuttal. The authors have addressed most of my concerns. The JL random projection baseline has been added and the heuristic method of reusing the eigenspace for some iterations has been explored, which I appreciate a lot. From the current experiments, the proposed method seems effective, though it will be more convincing if the method can be tested on larger models or harder datasets.

I think the topic of the paper is quite interesting and the idea of bounding $M_t - \Sigma_t$ uniformly is also interesting. The theory indeed manifests the effectiveness of the proposed method. As a result, I increase my point to 7.

---

> ### Author Response · Authors · 2020-11-25
> **Response to Reviewer 3**
>
> We thank the reviewer for the constructive feedback. We are encouraged that the reviewer appreciates our uniform bound by generic chaining techniques. We have revised our paper and did more experiments based on your suggestions. Here we explain in detail how we address your comments.
>
> 1. ‘’The establishment of (28) requires ...''
>
>  We have corrected those typos.
>
>
>
>
> 2. ''The necessity of using an extra dataset. If reducing dimensionality is the most important, why not just use a random projection.''
>
> The reviewer has raised a very interesting point. We have discussed in the introduction that one could potentially implement the projection through private subspace identification on the private dataset, i.e., private PCA. However, to our best knowledge, all existing methods on private PCA have reconstruction error scaling with $\sqrt{p}$ (Dwork et al., 2014), which will be propagated to the optimization error.
>
> We have also added Johnson-Linderstrauss (JL) random projection with DP-SGD as a baseline in Figures 3 and 4 by using a Gaussian random projector. We observed that this approach in general has a much lower accuracy than our proposed method. We think this is because random projection introduces a larger subspace reconstruction error due to the random projector than the injected noise error reduced by projection. Also, we acknowledge that JL random projection with non-private SGD does not provide a formal privacy guarantee. The private JL projection method in [1] requires further Gaussian noise added to the rectangular diagonal matrix under the singular value decomposition of the data matrix.
>
>
> 3. ''The proposed method has a huge space complexity and computation complexity. ''
>
> Thanks to the structure of our problem, our proposed method actually does not suffer huge space and computational complexity.  We use the power method Lanczos in our experiments. In Lanczos, we compute the matrix-vector product $A^T v$, where $A$ can be written as $\sum g_ig_i^T$ with individual stochastic gradient $g_i$. Thus, we do not have to explicitly store the matrix $A$. Also, the matrix-vector product can be decomposed into vector inner products:
> $$
> A^T v = \sum g_ig_i^Tv = \sum g_i(g_i^T v),
> $$
> which is a much faster computation than generic matrix-vector multiplication.
>
>
> 4. ''the difference between consecutive $w_t$ is so small that the difference between consecutive $V(w_t)$ is also small, implying we can reuse the eigenspace''.
>
> We appreciate the reviewer's input and we agree that it is a great idea to reuse the eigenspace if the subspaces are close for consecutive iterations. We have implemented this heuristic method, i.e., PDP-SGD that updates the projector every s = 1, 10, 20 iterations in our experiment and added results in Figure 6. Those results show that PDP-SGD with a reduced eigenspace computation also achieves good performance and improves the accuracy over DP-SGD for a certain privacy level.

---

### Official Review · AnonReviewer4 · 2020-10-27
**New private SGD method to improve the utility**

**Rating:** 6
**Confidence:** 3

**Review:**

The paper proposes a new private SGD method by projecting the noise gradient onto a subspace, which is estimated using some public datasets. The proposed method can improve the utility guarantee by reducing the dependence of the problem dimension p. The idea of the proposed method is interesting, but the assumptions in the current paper seem to be very strong, and the evaluations in the current paper are not convincing to show that the proposed method is beneficial.
1. It is not always true in practice that we can have access to the public dataset.
2. In Theorem 4, the assumption about the principal component of the gradient looks very strong.
3. How will the size of the public dataset affect the performance of the proposed method? It is important to have empirical evaluations on this.
4. Another issue of the proposed method seems to be the computational barrier. Why can the proposed method only deal with 10000 samples in MNIST dataset compared with DPSGD, which can deal with the whole training dataset?
5. I think the most valuable thing about the proposed method is to reduce the dependence of the problem dimension p. Therefore, it is very important for the authors to consider larger models such as resnet. If the proposed method can achieve better performance on larger models, I think the proposed method is very convincing.

-----
After reading the author response and other reviews, most of my concerns have been addressed. I would like to increase my score to 6. It would also be interesting to compare the performance of DP-SGD and PDP-SGD when $\epsilon$ is relatively large as suggested by Figure 7.

---

> ### Author Response · Authors · 2020-11-25
> **Response to Reviewer 4**
>
> We thank the reviewer for the insightful feedback. We are encouraged that the reviewer found our idea to be interesting. Here we explain how we address your concerns.
>
> 1. ‘’It is not always true in practice that we can have access to the public dataset.‘’
>
> There is a long line of work that studies private data analysis with access to an auxiliary public dataset, i.e., Bassily et al., 2019b; 2020; Feldman et al., 2018; Avent et al., 2017; Papernot et al., 2017. This setting interpolates between private learning (where all examples are private) and classical learning (where all examples are public). Bassily et al., 2019b provide several motivations to consider this setting: First, in practice, it is feasible to collect some amount of “public“ data that poses no privacy concerns. For example, in the language of consumer privacy, there is a considerable amount of data collected from the so-called “opt-in” users, who voluntarily offer or sell their data to companies or organizations. There are also a variety of other sources of public data that can be harnessed. For example, the data collected by the US Census prior to the 2020 differential privacy deployment are publicly available. Leveraging (small) auxiliary public data has been a promising approach to differential privacy, especially since it enables data analyses that are impossible with only private data (e.g., releasing statistical queries with unbounded Littlestone dimension (Bassily et al., 2020)).
>
>
> 2. ‘’In Theorem 4, the assumption about the principal component of the gradient looks very strong. ‘’
>
> The principal component of the gradient assumption is motivated and supported by the known observations in Li et al., 2020; Gur-Ari et al.,
> 2018 that, in an over-parameterized regime, the set of sample gradients along the training trajectory is often contained in the top principal subspace. We also provided our empirical evaluation of this structure in terms of the eigenvalues of the gradient second moments matrix in Figure 1, which shows that the eigenvalues decay very fast from the largest to smallest and only a few eigenvalues corresponding to the top principal component dominate. Thus, we believe it is reasonable to assume the principal component dominates.
>
> Besides, our main result in Theorem 4, i.e., the error rate of the $\ell_2$-norm of the principal component of the gradient holds without assuming the principal component dominates. We have explicitly stated in the paper that, with the additional assumption that the principal component of the gradient dominates, one can derive a further bound on the gradient norm.
>
>
> 3. ‘’How will the size of the public dataset affect the performance of the proposed method? It is important to have empirical evaluations on this.''
>
> As shown in the theoretical bound, i.e., with $m$ public samples, the subspace reconstruction error scales as $ \frac{1}{\sqrt{m}}$ ignoring other factors. This shows that the error will decrease as the size of the public dataset increases. We also added empirical evaluations in the paper (Figure 5 (b)) with different public dataset sizes. Due to the space limit, more results on this are added in the Appendix. Our results show that the train/test accuracy increases as public size increases from 50 to 150. We also added a discussion on this behavior that increasing $m$ helps to reduce the subspace reconstruction error as suggested by the theoretical bound.
>
>
> 4. ‘’Another issue of the proposed method seems to be the computational barrier. Why can the proposed method only deal with 10000 samples in MNIST dataset compared with DPSGD, which can deal with the whole training dataset?''
>
>
> We have also added results for MNIST with 20000 samples, 50000 samples, and Fashion MNIST with 50000 samples in the revision.
> Results have been added to the main paper in Figure 7. Due to the space limit, additional results are added in the Appendix. Those results show that the same story (PDP-SGD performs better than DP-SGD in the high privacy regime, where the privacy loss $\epsilon$ is small) observed in 10000 training samples also holds for other numbers of training samples.
>
> To deal with the computation, we explored a heuristic method suggested by Reviewer3 that we reuse the eigenspace for several iterations if the difference between consecutive subspaces is small.
> We implemented PDP-SGD that updates the projector every s = 1, 10, 20 iterations and we added the results in Figure 6 in the main paper. Those results show that PDP-SGD with a reduced eigenspace computation also improves the accuracy over DP-SGD for a certain privacy level.

---

### Official Review · AnonReviewer2 · 2020-10-29

**Rating:** 6
**Confidence:** 3

**Review:**

The author proposes an algorithm that improves DPSGD by using public data to identify a lower-dimensional space where the gradients lie in. They show that the algorithm can provide a better convergence guarantee, specifically, p reduced to log(p). They also conducted experiments to show the proposed algorithm outperforms the generic DPSGD especially at small epsilon with a small amount of public data.

The topic of the paper is interesting, and the presentation is clear. I hope the significance of the paper can be better justified. On the theory side, I think it needs more comparison with prior work and needs a better explanation of some of the key concepts. On the experiment side, to make the results more convincing, it might be better to conduct evaluations on harder datasets.

Comments:

- In related work, it is said that "Kairouz et al. (2020) .... In comparison, our work studies both convex and non-convex problems and our analysis applies for more general low-dimensional structures that can be characterized by small γ2 functions (Talagrand, 2014) (e.g., low-rank gradients and fast decay in the gradient coordinates)". I believe Kairouz et al was operating on private data only and that might be a major difference. And it might still be interesting to compare the results in more detail, e.g. how do the bounds compare on convex problems, how do the two measurements of dimensionality compare.

- Line 4 of Alg 1: better to say \tilde{z_i} \in S_h than enumerating i from 1 to m (m is undefined in the algorithm).

- It is mentioned that if M is evaluated on fresh public samples at each iteration, then the deviation of M from Σ can be analyzed easily. I wonder if it worth discussing what would happen if partition the public data (given that we have more than T samples) into T disjoint subsets for each iteration. Would the tradeoff be worse?

- Since γ(M, d) is the main component in the analysis and the bound, can you provide more intuition and explanation of it, like how large it can get, on what kind of dataset it might be small/large, would model architecture make a difference etc.?

- A question regarding the experiment setup: I didn't quite understand why a subset of 10000, instead of (60000 - 100), of the original data is picked for training. 10000 is definitely a legitimate setting, but I just wonder if there is a reason for picking that number and whether the number of examples has any influence on the results.

- In the experiments in Figure 5, as k=50 (or 70) outperforms DPSGD without projection, it might make more sense to further increase k to demonstrate the tradeoff between the two types of errors.

- A minor point: Sec 4 said "Papernot et al. (2020) shows the accuracy of DP-SGD is around 80% when ε ≈ 1", which I'm not sure is accurate. I didn't find Papernot et at (in Table 4) reporting the accuracy for epsilon around 1. If the statement is referring to the leftmost point in Fig 3 of Papernot et al., I believe it is not a fair comparison. The figure basically showed a training process in terms of epsilon instead of steps, so the result is not optimized for epsilon = 1 (but is rather for epsilon = 2.93). On the other hand, the [tensorflow privacy tutorial](https://github.com/tensorflow/privacy/tree/master/tutorials) achieves 95% at epsilon = 1.19.

---

> ### Author Response · Authors · 2020-11-25
> **Response to Reviewer 2**
>
> We thank the reviewer for the constructive suggestions on improving the quality of this paper. Here we explain how we revised our paper based on your comments.
>
> 1. Compare the results with Kairouz et al. (2020) in more detail.
>
> We have updated our statement to acknowledge that the major difference with Kairouz et al. (2020) on convex problems is that Kairouz et al. (2020) only operates on private data. Our result on convex problems is presented in Theorem 6 (Due to the space limit, Theorem 6 is deferred to the Appendix.), we have added a paragraph after Theorem 6 to explicitly discuss the comparison with Kairouz et al. (2020).
>
> 2. Line 4 of Alg 1: better to say $\tilde{z_i} \in S_h$ than enumerating i from 1 to m.
>
> We have revised this part based on your comment.
>
> 3. I wonder if it worth discussing what would happen if partition the public data (given that we have more than T samples) into T disjoint subsets for each iteration. Would the trade-off be worse?
>
> The reviewer has raised a very good point. We think it is definitely meaningful to discuss this scenario.  We have added a discussion on this data splitting method for dealing with the dependency after we present the Ahlswede-Winter Inequality (the first paragraph in Section 3.1). If splitting m samples into T disjoint subsets, one will have m/T public samples at every iteration and the deviation error will scale with $(\sqrt{T}/\sqrt{m})$. This will lead to a worse trade-off between the subspace construction error and optimization error due to the dependence on T.
>
> 4. More intuition and explanation of $\gamma(M,d)$.
>
> At a high level, $\gamma_2(M,d)$ is related to $\sqrt{\log N(M,d,\epsilon)}$ where $N(M,d,\epsilon)$ is the covering number of the set  $M$ with $\epsilon$ balls with metric $d$, but it is considerably sharper. Such sharpening has happened in two stages in the literature: first, based on chaining, which considers an integral over all $\epsilon$ yielding the Dudley bound, and subsequently, based on generic chaining, which considers a hierarchical covering, developed by Talagrand and colleagues, and which yields the sharpest bounds of this type. The official perspective of generic chaining is to view $\gamma_2(M,d)$ as an upper (and lower) bound on suprema of Gaussian processes indexed on $M$ and with metric $d$ [Theorem 2.4.1 in Talagrand, 2014].
> Considering $d$ to be the $\ell_2$ norm distance, $\gamma_2(M,d)$ will be the same order as the Gaussian width of $M$ [Vershynin, 2019], which is a scaled version of the mean width of $M$. Structured sets (of gradients) have small Gaussian widths, e.g., a $L_1$ unit ball in $\mathbb{R}^p$ has a Gaussian width of $O(\sqrt{\log p})$, wheras a $L_2$ unit ball in $\mathbb{R}^p$ has a Gaussian width of $O(\sqrt{p})$. Based on the composition properties, one can construct examples of the gradient spaces that is composed of different subsets.  If $M = M_1 + M_2 = \{ m_1 + m_2, m_1 \in M_1, m_2 \in M_2\}$,
> the Minkowski sum, then $\gamma_2(M,d) \leq c (\gamma_2(M_1,d) + \gamma_2(M_2,d))$ (Theorem 2.4.15 in [Talagrand, 2014]), where $c$ is an absolute constant. If $M$ is a union of several subset, i.e., $M = \cup_{h=1}^D M_{h}$,
> then by using an union bound, we have $\gamma_2(M,d) \leq \sqrt{\log D} \max_{h} \gamma_2(M_{h},d)$. Thus, if  $M$ is an union of $D = p^s$ ellipsoids, i.e., polynomial in $p$, then $\gamma_2(M,\| \cdot \|_2) \leq O\left( \sqrt{s}\log p\right)$.
> We have added a more detailed discussion in the Appendix.
>
>
> 5. The experiment setup: why a subset of 10000, whether the number of examples has any influence on the results.
>
> To study whether the number of examples has any influence on the results, we also added results for MNIST with 20000 samples, 50000 samples, and Fashion MNIST with 50000 samples in the revision (The 100 public samples are randomly sampled from the rest 10000 samples.). Those results show that the same story (PDP-SGD performs better than DP-SGD for small $\epsilon$ regime) observed in 10000 training samples also holds for other numbers of training samples. Note that a smaller training sample size imposes more challenges on private training.
>
> 6. In the experiments in Figure 5, as k=50 (or 70) outperforms DPSGD without projection, it might make more sense to further increase k to demonstrate the tradeoff between the two types of errors.
>
> We agree that it could be an interesting idea to explore different values of $k$. Our choice of $k$ (between 10 to 50) is motivated by the empirical low-rank structure in the stochastic gradients (see Figure 1). Similar structures have been observed in Papyan (2019) as well.
>
> 7. ``A minor point: Sec 4 said "Papernot et al. (2020) shows the accuracy of DP-SGD is around 80 ...
>
> We thank the reviewer for pointing out this. We have revised our discussion in Sec 4 and we acknowledge that the result in Papernot et al. (2020) shows training dynamics in terms of privacy loss $\epsilon$ instead of epochs.

---

### Author Response · Authors · 2020-11-25
**Common Response-Summary of Updates**

We thank all reviewers for your constructive suggestions and comments, which indeed help us improve the quality of this paper. We have submitted an updated version based on these comments and included additional experiments. Here, we summarize the updates of the revision.

1) As suggested by Reviewer3, we added Johnson-Linderstrauss (JL) random projection with DP-SGD as a baseline in Figures 3 and 4 by using a Gaussian random projector.

2) We explored a heuristic method suggested by Reviewer3 that we can reuse the eigenspace for some iterations to save the computation. Thus, we implemented PDP-SGD that updates the projector every s = 1, 10, 20 iterations, and we added the results in Figure 6 in the main paper. More results are deferred to the Appendix due to the space limitation. Those results show that PDP-SGD with a reduced eigenspace computation also achieves good performance and improves the accuracy over DP-SGD for a certain privacy level.

3) As suggested by Reviewer2,
to study whether the number of examples has any influence on the results, we added results for MNIST with 20000 samples, 50000 samples, and Fashion MNIST with 50000 samples in the main paper and the Appendix. We observed that PDP-SGD performs better than DP-SGD in the high privacy regime, which corresponds to small privacy loss $\epsilon$. This is consistent with the results we reported using 10000 training samples.

4) As suggested by Reviewer4, we added experiments with different sizes of public datasets to empirically study the effect of public data size.
The results are included in Figure 5(b) and additional results are added in the Appendix due to space limitation.

5) As suggested by  Reviewer2, we added a discussion on the sample splitting method for bypassing the dependency issue with sample reuse in the first paragraph in Section 3.

6) We have corrected typos and added more discussion with prior work.

---

### Decision · Program_Chairs · 2021-01-07
**Final Decision**

**Decision:**

Accept (Poster)

**Comment:**

Based on the observation that the stochastic gradients for deep nets often stay in a low dimensional subspace, this paper proposes projected differential private SGD (DP-SGD) that performs noise reduction by projecting the noisy gradients to a low-dimensional subspace. Under certain assumptions, the authors provide a theoretical analysis and empirical evaluations to show that the proposed algorithm can substantially improve the accuracy of DP-SGD in the high privacy regime. There is unanimous support to accept this paper after the author’s response. Thus, I recommend accept.